# Static charge is an ionic molecular fragment

Yan Fang[1,2,3], Chi Kit Ao[1,3], Yan Jiang[1], Yajuan Sun[1], Linfeng Chen [1] & Siowling Soh [1] ✉

What is static charge? Despite the long history of research, the identity of static charge and mechanism by which static is generated by contact electrification are still unknown. Investigations are challenging due to the complexity of surfaces. This study involves the molecular-scale analysis of contact electrification using highly well-defined surfaces functionalized with a self-assembled monolayer of alkylsilanes. Analyses show the elementary molecular steps of contact electrification: the exact location of heterolytic cleavage of covalent bonds (i.e., Si-C bond), exact charged species generated (i.e., alkyl carbocation), and transfer of molecular fragments. The strong correlation between charge generation and molecular fragments due to their signature odd-even effects further shows that contact electrification is based on cleavage of covalent bonds and transfer of ionic molecular fragments. Static charge is thus an alkyl carbocation; in general, it is an ionic molecular fragment. This mechanism based on cleavage of covalent bonds is applicable to general types of insulating materials, such as covalently bonded polymers. The odd-even effect of charging caused by the difference of only one atom explains the highly sensitive nature of contact electrification.

Static charge is the immobile charged species on an insulating surface. Beyond knowing its existence, however, it is not known what the species of static charge is despite having been studied for more than 2000 years[1,2]. Current science typically only considers a static charge as a conceptual "point charge" without any information of what its chemical identity is. Static charge is generated whenever two surfaces are brought into contact and are then separated. This process of contact electrification, including simple contacts between two surfaces, typically generates large amounts of static charge (e.g., 1–10 μC/m²) on almost all types of insulating materials, including polymers, rubbers, textiles, and inorganic materials. The mechanism of contact electrification, however, is not known. Without understanding how static charge is generated, it has been difficult to identify the chemical species of static charge.

Static is tremendously important in our lives. It is widely felt directly (e.g., the electric shock when touching a doorknob), observed (e.g., lightning), experienced (e.g., static in clothes after drying), studied (e.g., including in elementary science), and demonstrated (e.g., the raising of hair when touching a Van de Graaff generator). Static

charge is the fundamental element of electrostatics. Static charge generated by contact electrification is the driving force of many operations based on electrostatic that is used in a large number of applications, including electrophotography, separation[3], self-assembly[4], electrochemistry[5], sensors[6], microfluidics[3], adhesion[7], powering of wearable electronics[8], and harvesting of energy from nature (e.g., rain, wind, or waves)[9]. On the other hand, excessive amounts of static charge generated by contact electrification give rise to a wide range of severe problems in industry, such as causing explosion of flammable substances, damaging equipment (e.g., electronics), reducing quality of products (e.g., drugs), and decreasing efficiency of manufacturing processes (e.g., blockage of pipes due to fouling)[10–12]. Despite its importance, the general characteristics and behaviors of static are very poorly understood[1,13]. In contrast to the well-understood theoretical electrostatics (e.g., calculation of electric fields), not much is known about electrostatics in practice, including wide-ranging issues from the dynamics of static charge to the dependence of static on environmental conditions (e.g., temperature or humidity). This lack of understanding of electrostatics in practice has

[1]Department of Chemical and Biomolecular Engineering, National University of Singapore, 4 Engineering Drive 4, Singapore 117585, Singapore. [2]College of Biotechnology and Pharmaceutical Engineering, State Key Laboratory of Materials-Oriented Chemical Engineering, Nanjing Tech University, 30# Puzhu South Road, Nanjing 211816, China. [3]These authors contributed equally: Yan Fang, Chi Kit Ao. ✉e-mail: chessl@nus.edu.sg

greatly hindered the development of technologies in this field. The issue is that research in experimental electrostatics has been driven almost entirely by empirical data via extensive experimental testing of very specific conditions. This approach has led to numerous accounts of conflicting and anomalous results reported in literature. Hence, it is critically needed to understand the fundamentals of contact electrification for developing further general understanding of the characteristics and behaviors of static. Two most important fundamental issues need to be addressed: what is the identity of static charge and how is static generated by contact electrification.

The study of the mechanism of contact electrification generally relies on one reproducible result of contact electrification: the contact of two initially uncharged surfaces generates one surface that is positively charged and another surface that is negatively charged. Hence, the process involves the transfer of a charged species from one surface to the other during contact. Several mechanisms for the transfer of charge have been proposed in previous studies for explaining contact electrification; however, there are strong doubts about the fundamental plausibility of the proposed mechanisms.

Two mechanisms commonly debated in literature are the transfer of electrons[5,14] and transfer of ions[15]. Electron transfer is often not regarded as fundamentally possible because it is not energetically favorable for electrons to be transferred between insulators during simple contacts (e.g., removing electrons from covalently bonded atoms with full valence shells requires large amounts of energy) and charge generation does not correlate with electronic properties of insulating materials[15]. For ion transfer, the mechanism has only been shown using surfaces that are prepared specifically to contain mobile ions[16]. However, a vast majority of insulating surfaces do not contain mobile ions for ion transfer, including common types of polymers that tend to charge easily by contact electrification (e.g., polytetrafluoroethylene). One proposed source of mobile ions is based on water adsorbed onto surfaces due to the humidity of the surrounding atmosphere (e.g., aqueous hydroxide and hydronium ions)[15]; however, other studies have found that contact electrification occurred even without the presence of water[17]. A third mechanism that has been discussed to a much lesser extent involves the transfer of charged materials (i.e., matter defined as nanoscale or larger). Previous studies have found that materials transferred between surfaces after contact[18–21]. Other studies found that the amount of charge generated correlated with softness[19,22–27], adhesiveness[26,28,29], glass transition of polymers[22], thermal history[23], roughness[30,31], and concentration of fillers[24] of the contacting materials. However, material transfer is generally not considered to be the mechanism of contact electrification due to the uncertainty of the fundamental concept; in particular, it is currently unknown whether the materials that transferred from one surface to another are charged. The limited amount of different pieces of materials (i.e., nanoscale or larger) transferred may not account for the significant amount of charge generated by contact electrification[32]. In general, there is a severe lack of evidence that supports any of these mechanisms because the results reported are based on indirect and macroscale observations.

Investigations of the fundamental mechanism of contact electrification have been extremely challenging because the chemical and physical properties of insulating surfaces are largely unclear, complex, and difficult to characterize. Surfaces usually have a broad variety of ill-defined surface defects, dangling bonds, chemical groups, and surface topology. Because properties of surfaces are mostly unclear, discussions of results involve the consideration of many different possible explanations instead of a definite conclusion[2,11,33]. These issues are especially severe for polymers, which tend to charge highly and have many applications based on static. In addition, the amounts of static charge generated by contact electrification are extremely small (e.g., only a femtomolar of charged species is produced by a typical charge density of $1 \mu C/m^2$ on a surface of $1 cm^2$). Because of these complications, it is generally perceived that the phenomenon "may never be predictable"[1,34].

This manuscript describes the molecular-scale investigation of the fundamental mechanism of contact electrification using surfaces that are chemically and physically well-defined for studying the phenomenon clearly and highly sensitive analytical equipment for analyzing surfaces. Our experiments first involve coating self-assembled monolayers (SAMs) of alkylsilanes onto surfaces of mica (see Methods section for more details on the materials and methods). We prepare eight different types of alkylsilane-coated surfaces; each surface is coated with a type of alkylsilanes of a specific number of carbon atoms, $n$, from 1 to 8 (Fig. 1a). These surfaces are well-defined. Through coating the SAMs using solution-phase silanization[35], highly ordered structures of the molecules on the surface are produced[36]. The surfaces have only one type of functional group—the alkyl groups—at the topmost surface for contact electrification; thus, the analyses are not complicated by other types of functional groups. Physically, mica is atomically smooth[37]. We measured that mica had a surface roughness of <0.2 nm (Table S2). After coating the alkylsilanes, the surfaces were still extremely smooth with surface roughnesses of <0.6 nm.

For studying the mechanism of contact electrification, we contact-charge the alkylsilane-coated surface against the bare (i.e., uncoated) surface of mica (Fig. 1b, c). These two well-defined but different types of surfaces enable the results of charge transfer from one surface to another to be clearly interpreted. Analysis of the well-defined surfaces after contact allows us to understand the elementary molecular mechanism of contact electrification in a step-by-step manner.

These surfaces are representative for studying contact electrification of general types of insulating materials because the alkylsilanes are covalently bonded—this type of chemical bonding is commonly found in insulating materials, especially polymers. In addition, the alkyl group is the most common type of functional group found in many insulating materials, such as polymers.

## Results
### Contact electrification of well-defined surfaces
Our experiments involved first coating the SAMs on the surface of mica and analyzing the surfaces. We verified that the surfaces were successfully coated with the alkylsilanes by detecting the presence of the molecular fragment $-O_3-Si-(CH_2)_{n-1}-CH_3$ via the time-of-flight secondary ion mass spectrometry (ToF-SIMS; Fig. S1) and substantial increase in carbon compared to the uncoated mica by X-ray photoelectron spectroscopy (XPS; Table S1).

We performed the contact electrification by bringing the alkylsilane-coated surface of a specific $n$ (i.e., $C_n$-SAM) into contact with the uncoated surface of mica (i.e., $C_n$-mica) twenty times under ambient conditions (i.e., humidity ~60%). After contact, the charges of both the surfaces were measured immediately (i.e., <2 s) using a Faraday cup connected to an electrometer (Keithley 6514). The experiment was performed for the eight types of $C_n$-SAM and $C_n$-mica with $n$ from 1 to 8.

Results showed that the amount of charge generated on both surfaces increased, in general, with increasing $n$ (Fig. 2). The increase was large. The amount of charge generated when $n = 8$ compared to $n = 1$ was around eleven times for the alkylsilane-coated surfaces and six times for the uncoated surfaces when a typical pressure of ~100 Pa was used for contacting the surfaces ("Normal pressure" in Fig. 2a). Importantly, the increase was not monotonic and followed a remarkable trend: both the contacting surfaces charged higher when the alkylsilanes coated onto the surfaces had even numbers of carbon atoms, $n_{even}$, than odd numbers of carbon atoms, $n_{odd}$ (i.e., an odd-even effect). For example, the negative charge of the alkylsilane-coated surface and positive charge of the uncoated surface were larger when $n = 2$ compared to when $n = 1$ or $n = 3$. This significant and systematic odd-even effect was not initially expected to occur due to the highly

stochastic and complex nature of contact electrification. The phenomenon was general. Similar trends were observed when lower amounts of pressure were used for contacting the materials ("Low pressure" of ~60 Pa and "Ultra-low pressure" of ~15 Pa in Fig. 2a). The alkylsilane-coated surfaces exhibited an anomalous charging behavior over time but maintained the odd-even effect (Fig. S2). In general, these results showed that the amount of charge generated depended strongly on the specific value of $n$−hence, the charging by contact electrification was because of the alkylsilanes.

For understanding the role of the alkylsilanes, we analyzed the surface roughness of the uncoated surfaces, $C_n$-mica, after contact electrification with $C_n$-SAM for all $n$ from 1 to 8 by Atomic Force Microscopy (AFM). The atomically smooth surface of the uncoated mica was the ideal surface for studying any changes in surface topology. Results showed that the surface roughnesses of the $C_n$-mica of all $n$ from 1 to 8 were >2 nm; hence, they were much rougher after than before contact (Fig. 2b). As a control experiment, we contact-charged two uncoated surfaces and found that their surface roughnesses were still very low at around 1 nm. These results thus indicated that

molecular fragments of the monolayers of alkylsilanes transferred from the alkylsilane-coated surface onto the uncoated surface during contact electrification.

The results showed that the surface roughness of the uncoated surfaces increased, in general, with increasing $n$ after contact electrification and exhibited the remarkable odd-even effect. There was thus a strong correlation between the amount of charge generated (Fig. 2a) and surface roughness (Fig. 2b) of the uncoated surfaces after contact electrification, especially when they both exhibited the unique signature characteristic of the odd-even effect. This unique correlation indicated that the charge generated was based on the transfer of the molecular fragments of the alkylsilanes.

**Analyzing molecular fragments after contact electrification**
We investigated the specific types of molecular fragments that transferred by analyzing the major differences in the chemical composition of the uncoated surfaces before and after contact electrification against the alkylsilane-coated surfaces (Fig. 3). The major difference detected by XPS was clearly observed from the high-resolution

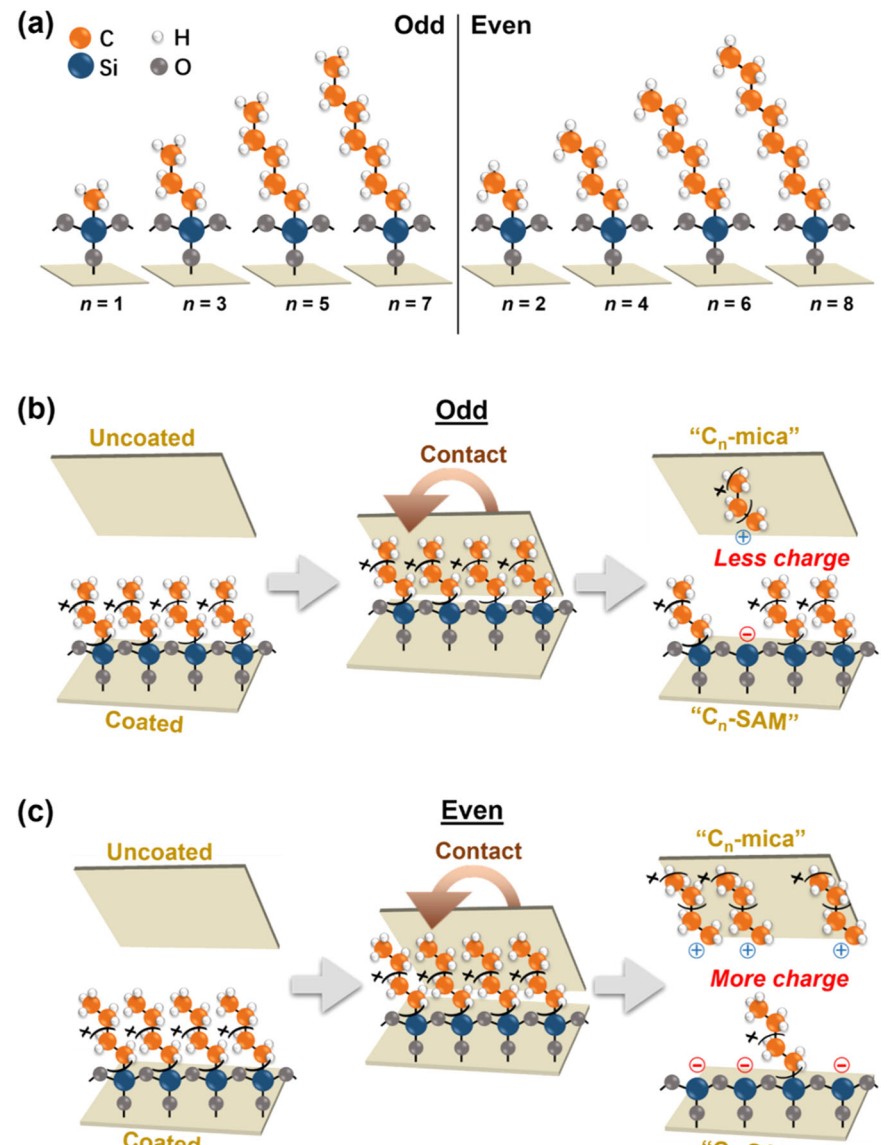

**Fig. 1 | Contact electrification between mica coated with a self-assembled monolayer (SAM) of alkylsilanes and uncoated surface of mica. a** Eight types of alkylsilanes with different numbers of carbon atoms, $n$ = 1 to 8, were used to coat mica. Schemes illustrate the mechanism of contact electrification between an uncoated surface of mica and a surface coated with alkylsilanes of **b** an odd number or **c** an even number of carbon atoms.

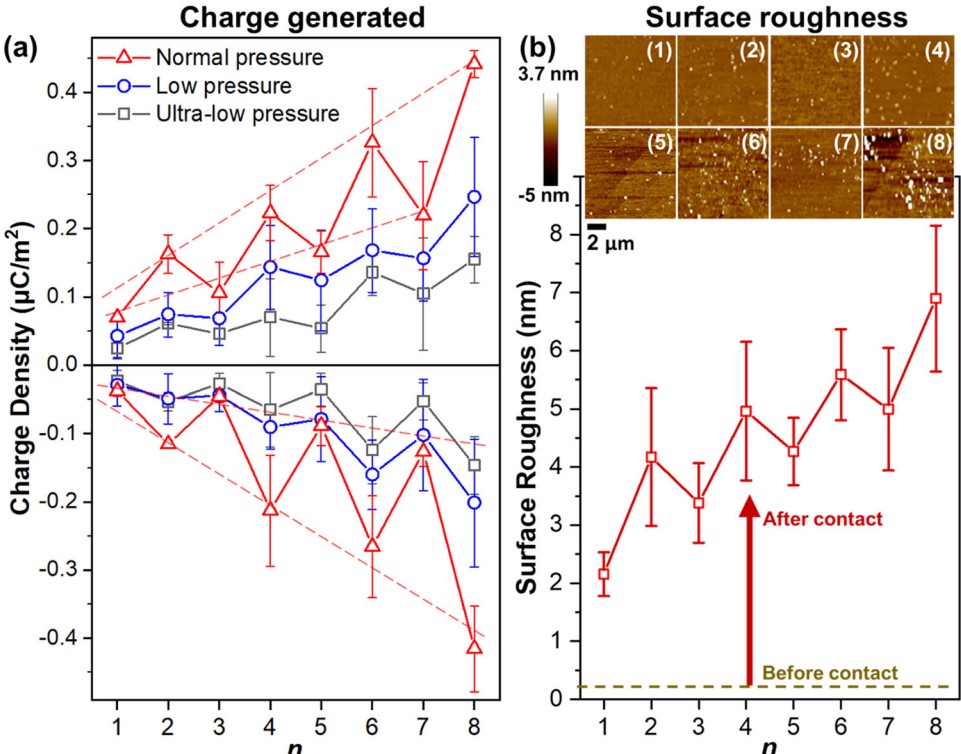

**Fig. 2 | Strong correlation between charge and surface roughness generated by contact electrification. a** Charge generated by the contact electrification between the uncoated surface (i.e., positive data shown in plot on top) and coated surface with alkylsilanes of $n$ number of carbon atoms (i.e., negative data shown in plot at the bottom). **b** Root-mean-square (RMS) surface roughness of the uncoated surface before (dotted yellow line) and after (solid red lines with squares) contact electrification with surface coated with alkylsilanes of $n$ number of carbon atoms. Insets show the atomic force microscopy (AFM) images of the respective surfaces (labeled with $n$). Data are from samples performed in triplicates and are plotted as mean ± standard deviation (SD). Source data are provided as a Source Data file.

spectrum of the C 1$s$. Before contact electrification of the uncoated mica, the C 1$s$ showed only one peak at 284.6 ± 0.1 eV that corresponded to the C-H/C-C ("Mica before contact" in Fig. 3b). After contact electrification, however, analyses of the uncoated surfaces, C$_n$-mica, of all $n$ from 1 to 8 showed the distinctive shoulder that was fitted by a new peak at 286.5 ± 0.1 eV due to the presence of C-OH.

Analysis by ToF-SIMS of the uncoated surface before contact electrification showed that the intensities at the values of $m/z$ that corresponded to the alcohols, $C_nH_{2n+1}OH$, of all $n$ from 1 to 8 were negligible (plots in yellow in Fig. 3c). After contact electrification of the uncoated surface, C$_n$-mica, of a specific $n$, the peak at the value of $m/z$ that corresponded to $C_nH_{2n+1}OH$ of the same $n$ appeared (plots in red in Fig. 3c). The appearance of the peak that corresponded to $C_nH_{2n+1}OH$ of the specific $n$ occurred for C$_n$-mica of all $n$ from 1 to 8. The intensities of the peaks were very high. When compared to the intensities of all the other values of $m/z$ of the full spectra analyzed by ToF-SIMS, these peaks were far stronger—they were either the highest or among the highest (i.e., besides the typical peaks expected of mica; red in Fig. 3d). Furthermore, these peaks showed the remarkable odd-even effect as well (Fig. 3e). Hence, these results showed that the alcohol, $C_nH_{2n+1}OH$, of the specific $n$ was not present initially on the uncoated surface before contact electrification and became present in large quantities after contact electrification.

Alcohol is the indicator of the formation of alkyl carbocation. Alkyl carbocations react spontaneously and rapidly with water due to the humidity (i.e., ~60%) of the surrounding atmosphere under normal ambient conditions to form alcohols and H$^+$ ions (e.g., the essential step in the classic reaction of hydration of alkenes; Fig. 3a)[38–40]. Therefore, the large quantities of alcohols produced indicated that alkyl carbocations were generated during contact electrification. ToF-SIMS spectra of the uncoated surfaces after contact electrification

against the alkylsilane-coated surfaces showed significant peaks at the values of $m/z$ that corresponded to the alkyl carbocations, $C_nH_{2n+1}^+$, of all values of $n$ compared to the intensities at other values of $m/z$ (Fig. S3); however, the peaks were not greatly higher than the background signal. Alkyl carbocations are highly unstable and reactive with extremely short lifetimes (e.g., nanoseconds and picoseconds)[41]; hence, they may not be detectable.

Importantly, the analyses by ToF-SIMS of the uncoated surfaces after contact electrification, C$_n$-mica, of all $n$ from 1 to 8 showed another set of peaks with very high intensities: the peaks with values of $m/z$ that corresponded to alkanes, $C_{2n}H_{4n+2}$, that had $2n$ number of carbon atoms (blue in Fig. 3d). These peaks were similarly very high when compared to other values of $m/z$ across the entire spectra. For example, an abundant amount of $C_2H_6$ at $m/z = 30$ was found on C$_1$-mica after contact electrification with C$_1$-SAM that was coated with the alkylsilane, -O$_3$-Si-CH$_3$. Importantly, the intensities at the values of $m/z$ that corresponded to $C_{2n}H_{4n+2}$ of all $n$ from 1 to 8 were negligible on the uncoated surface before contact electrification—these strong peaks appeared only after contact electrification (Fig. 4b). This result was especially significant because larger molecules were more susceptible to be broken down into smaller fragments by ToF-SIMS; hence, the intensities of the large molecules such as the alkanes with $2n$ number of carbon atoms should technically be small.

Alkanes with $2n$ number of carbon atoms, $C_{2n}H_{4n+2}$, can be generated readily by the termination of the alkyl radical, $C_nH_{2n+1}•$, with the same alkyl radical, $C_nH_{2n+1}•$, with $n$ number of carbon atoms. Hence, the presence of the abundant amounts of $C_{2n}H_{4n+2}$ strongly suggested that there was first homolytic cleavage at the Si-C bond of the alkylsilanes during contact electrification for forming the alkyl radical, $C_nH_{2n+1}•$, with $n$ number of carbon atoms and then the self-reaction of the alkyl radicals for forming the alkanes with $2n$ number of carbon

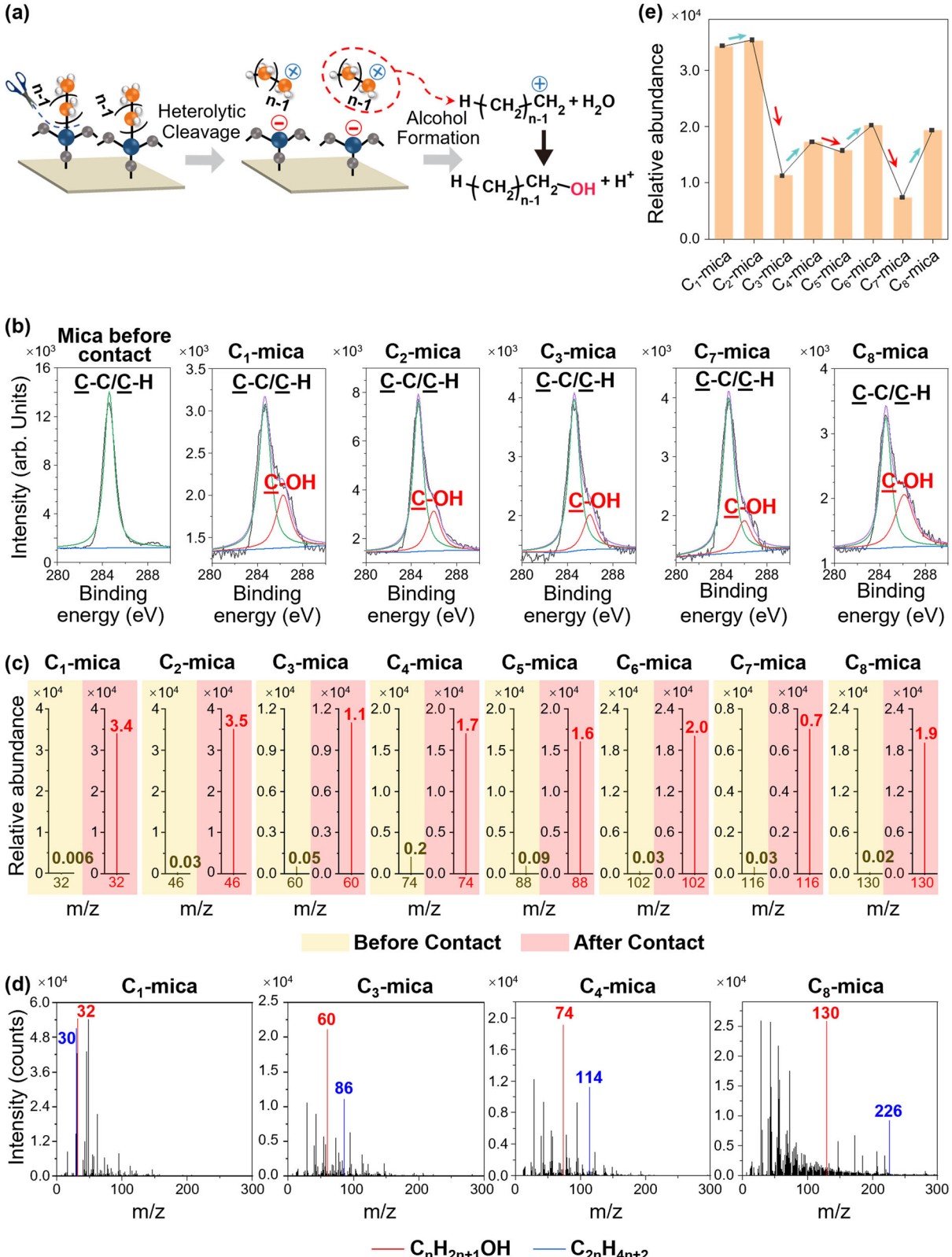

**Fig. 3 | Analyzing heterolytic cleavage by contact electrification. a** Reaction scheme of the heterolytic cleavage of alkylsilane that generates the alkyl carbocation. Reaction of alkyl carbocation with water (e.g., moisture from environment) produces alcohol. **b** X-ray photoelectron spectroscopy (XPS) spectra of the uncoated surfaces of mica that show the appearance of the C-OH group after contact electrification. The notation $C_n$-mica refers to the uncoated surface after contact electrification against the surface coated with alkylsilanes of $n$ number of carbon atoms. **c** Time-of-flight secondary ion mass spectrometry (ToF-SIMS) spectra of the alcohol ($C_nH_{2n+1}OH$) of the uncoated surfaces before (yellow) and after (red) contact electrification against the alkylsilane-coated surfaces of different $n$. **d** Full spectra of ToF-SIMS of the uncoated surfaces after contact electrification. The peaks that correspond to the typical elements of mica ($^{23}$Na, $^{27}$Al, $^{28}$Si, $^{39}$K, $^{41}$K) have been removed. **e** Odd-even effect of the ToF-SIMS intensities of the peaks of the alcohols ($C_nH_{2n+1}OH$) of the uncoated surfaces after contact electrification. Source data are provided as a Source Data file.

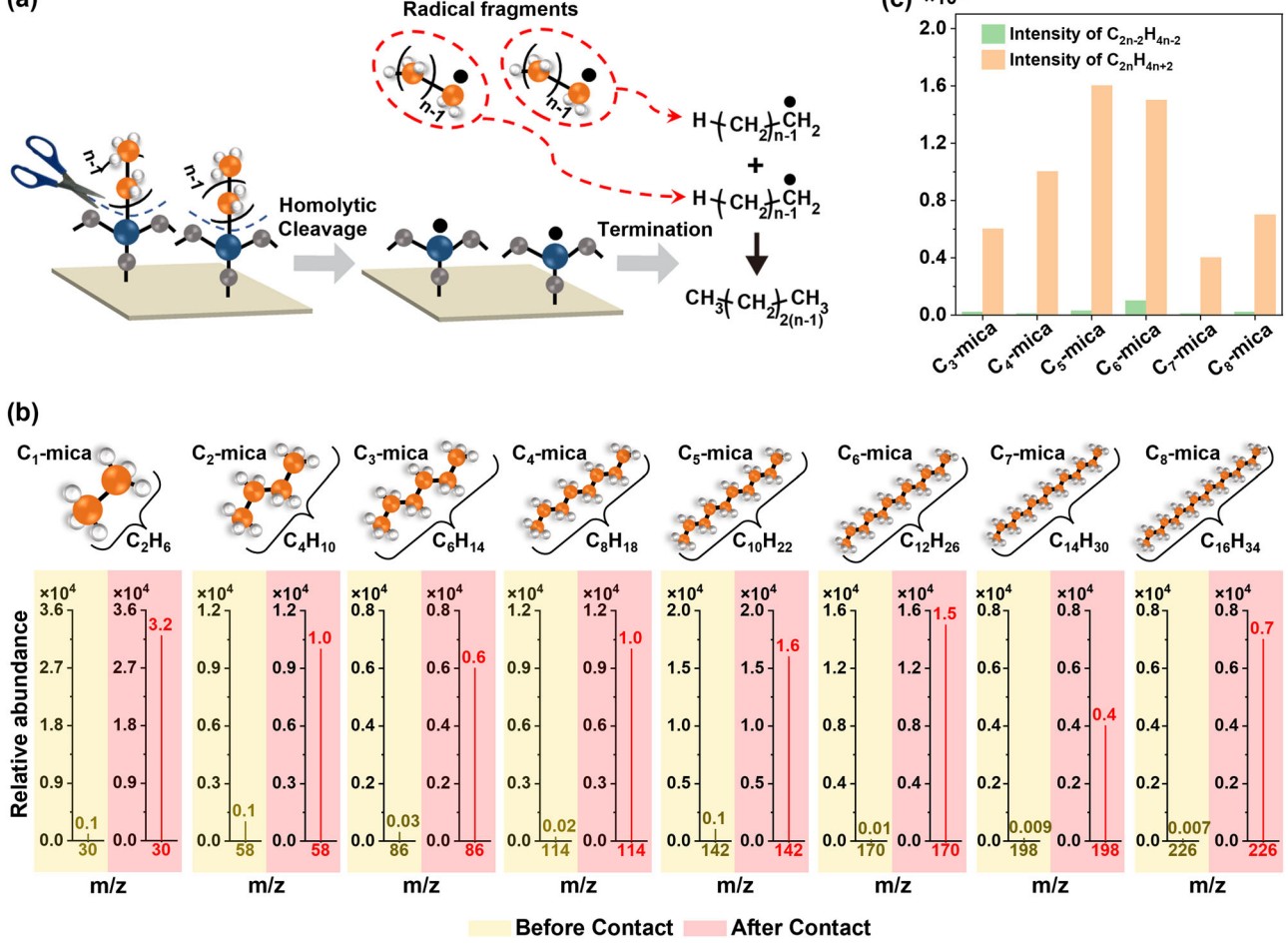

**Fig. 4 | Analyzing homolytic cleavage for identifying the position of bond cleavage by contact electrification along the alkylsilane functionalized on the surface. a** Reaction scheme of homolytic cleavage of alkylsilane that generates alkyl radicals. Self-reaction of alkyl radicals with $n$ number of carbon atoms produces an alkane with $2n$ number of carbon atoms. **b** ToF-SIMS spectra of the alkanes with $2n$ number of carbon atoms ($C_{2n}H_{4n+2}$) on the uncoated surfaces before (yellow) and after (red) contact electrification with the alkylsilane-coated surfaces of different $n$. **c** Comparing the ToF-SIMS intensities of the peaks of the alkanes with the $2n$ number of carbon atoms ($C_{2n}H_{4n+2}$) and the peaks of the alkanes with the $2(n-1)$ number of carbon atoms ($C_{2n-2}H_{4n-2}$) on the uncoated surfaces after contact electrification. Source data are provided as a Source Data file.

atoms (Fig. 4a). The presence of the abundant quantities of the alkanes, $C_{2n}H_{4n+2}$, and alcohols, $C_nH_{2n+1}OH$, thus enabled us to identify the location of bond cleavage to be at the Si-C bond.

We examined the possibility of cleavage at the C-C bonds of longer alkylsilanes (i.e., for $n > 2$). If the homolytic cleavage at Si-C bond formed the alkanes with $2n$ number of carbon atoms (i.e., $C_{2n}H_{4n+2}$), the homolytic cleavage at the first C-C bond next to the Si-C bond would form the alkanes with $2(n-1)$ number of carbon atoms (i.e., $C_{2(n-1)}H_{4(n-1)+2}$). Results from ToF-SIMS showed however that the peaks of alkanes with $2n$ number of carbon atoms were far stronger than the peaks of alkanes with $2(n-1)$ number of carbon atoms (Fig. 4c). These results thus indicated that the location of bond cleavage occurred mainly at the Si-C bond of the alkylsilanes on the coated surface. Analysis of the alkylsilane-coated surface by ToF-SIMS also showed that the location of the cleavage occurred at the Si-C bond (Fig. S4).

**Elementary steps of the mechanism of contact electrification**
For the odd-even effect, previous studies that examined the molecular structure of the self-assembled monolayer of alkyl chains found that the orientation of the terminal $CH_3$-$CH_2$- moiety at the outermost portion of the molecule can be normal or tilted away from the normal to the surface depending on whether the molecule is $n_{even}$ or $n_{odd}$.

When the terminal $CH_3$-$CH_2$- moiety is normal to the surface, the outermost portion of the surface is composed of mainly the methyl groups; when the $CH_3$-$CH_2$- moiety is tilted away from the normal, the outermost portion is composed of both the methyl groups and methylene groups. This odd-even molecular orientation can be determined by analyzing the wettability of liquids on the surfaces[35]. Specifically, the contact angles of different types of liquids (e.g., $n$-hexadecane and water) are lower when the terminal $CH_3$-$CH_2$- moiety is tilted away from the normal than when it is normal to the surface[42,43]. We measured the contact angles of liquids on the alkylsilane-coated surfaces. Results showed that the contact angles of water (Fig. S5a) and hexadecane for $n < 5$ (Fig. S5b) were generally lower for $n_{even}$ than for $n_{odd}$. This result thus indicated that the terminal $CH_3$-$CH_2$- moiety was tilted away from the normal for $n_{even}$ and directed toward the normal for $n_{odd}$ as illustrated in Fig. 1a–c.

Previous studies have reported that the frictional force between the contacting surfaces is larger when the terminal $CH_3$-$CH_2$- moiety of the self-assembled monolayer is tilted away from the normal[44]. The reason is because the presence of both the methyl groups and methylene groups when the terminal $CH_3$-$CH_2$- moiety is tilted away from the normal allows more atoms per unit area to be in contact between the surfaces than when only the methyl groups are present. When more atoms per unit area are in contact, the surfaces experience

larger Van der Waals forces and a larger frictional force. In our case, this result indicates that surfaces with $n_{even}$ experienced a larger frictional force than surfaces with $n_{odd}$. Hence, surfaces with $n_{even}$ have a larger amount of cleavage of bonds than surfaces with $n_{odd}$.

All these results enabled us to determine the elementary steps of the molecular mechanism of contact electrification. When the surfaces are brought into contact, the frictional force between the contacting surfaces causes the heterolytic cleavage of the alkylsilanes functionalized on the surface—specifically, the heterolytic cleavage occurs at the polar Si-C bond. This cleavage produces a -$O_3$-Si moiety that remains attached onto the surface and a free alkyl fragment. Based on the Sanderson's principle of electronegativity equalization, the group electronegativity of the -$O_3$-Si moiety is calculated to be higher than that of the alkyl fragment (see Methods for the calculation). Hence, the heterolytic cleavage produces a negatively charged -$O_3$-Si moiety on the surface and a free alkyl carbocation. The alkyl carbocation transfers from the alkylsilane-coated surface to the uncoated surface during contact. Surfaces of mica are reported to have the ability to adsorb many different types of molecules and ions via intermolecular forces, such as van der Waals forces[45,46] and electrostatic forces (i.e., due to the typically negatively charged surface of mica[47,48]). After separating the surfaces, the alkylsilane-coated surface charges negatively, whereas the uncoated surface charges positively due to the presence of the alkyl carbocations. The surface roughness of the uncoated surface increases due to the transfer of the ionic molecular fragments (i.e., alkyl carbocations). Because surfaces with $n_{even}$ experience larger frictional forces than surfaces with $n_{odd}$ during contact, there is more heterolytic cleavage of covalent bonds. Hence, more ionic molecular fragments are transferred and more charge is separated when the contacting surfaces had $n_{even}$ than $n_{odd}$ (Figs. 1b, c). This mechanism thus gives rise to the strong correlation between charge generation and surface roughness. Importantly, this molecular mechanism indicated that the static charge generated by contact electrification is an alkyl carbocation.

The alkyl carbocation then reacts spontaneously to become an alcohol, thus leaving the $H^+$ ion behind as the charged species on the surface (e.g., by remaining adsorbed on the surface of mica via van der Waals forces and/or hydrogen bonds). On the other hand, alkyl radicals produced by homolytic cleavage of bonds react via many different reaction pathways to form a wide range of products under normal ambient conditions, but not alcohols unless under specific conditions[49]. Previous studies that investigated reactions of alkyl radicals have not detected alcohols as the products[50]. In addition, the intensities of the species that are typically involved or produced in the radical reactions (i.e., $R_{n-1}CHO$, $R_nO_2\bullet$, $R_nO\bullet$, $R_nOOH$, $R_nONO_2$, $R_nOONO_2$, and $R_nOOR_n$) are found to be mostly negligible from our ToF-SIMS analyses.

## Discussion

We used well-defined surfaces for analyzing the mechanism of contact electrification at the molecular level. Clear results were obtained because the chemical and physical properties of the surfaces were well understood, without unknown factors that would complicate the analysis. We showed the elementary steps of contact electrification, including the exact location of cleavage of the covalent bond, exact charged molecular species generated (i.e., alkyl carbocations), and actual transfer of the charged molecular species for charge separation. The clarity at the molecular level indicated clearly that contact electrification occurs by first the heterolytic cleavage of covalent bonds and then transfer of the ionic molecular fragments. Besides showing that it occurs, we showed that this specific mechanism occurs in abundance—at quantities that correspond to the substantial amount of charge separation generated by contact electrification. Three results showed that the mechanism occurred in abundance. First, there is a strong correlation between the amount of charge generated (Fig. 2a)

and surface roughness (Fig. 2b) (i.e., both showed the general increasing trend and signature odd-even effect); changes in surface roughness are indicative of large-scale transfer of molecules. Second, there is a correlation between the amount of charge generated (Fig. 2a) and the transfer of molecules detected by ToF-SIMS (Fig. 3e) (i.e., both showed the unique odd-even effect). Third, large amounts of transfer of alcohols are detected by ToF-SIMS (Fig. 3). Previous studies have only considered the transfer of other types of species, such as electrons, mobile ions, and materials (i.e., nanoscale or larger), but not an ionic molecular fragment.

This mechanism is applicable to general types of insulating materials. Most insulating materials are covalently bonded. Hence, similar cleavage of covalent bonds will occur on the surfaces of other types of insulating materials according to the mechanism described in this study (e.g., similar cleavage of carbon-heteroatom or carbon-carbon bonds on polymeric surfaces and covalent bonds in inorganic materials). An important consideration is that surfaces are truncations of bulk 3D materials. Hence, surfaces have molecular groups that are largely less bonded covalently than within the bulk and typically have a vast variety of functionalization that produces groups that are less bonded (e.g., including groups with only one covalent bond on the surface) than within the bulk (e.g., the spontaneous functionalization of -OH groups by atmospheric moisture on inorganic materials, such as oxides). In our case, we found that the static charge is an alkyl carbocation. According to the elementary steps of this mechanism, the actual charged species generated by other types of insulating materials will depend on the types of chemical groups present on the surface—in general, static charge on insulating materials is an ionic molecular fragment.

This study showed for the first time that structural changes of molecules significantly affect the amount of charge generated by contact electrification—even by the difference of one atom (i.e., shown by the odd-even effect of charging). Factors previously reported to affect charging involve significant changes in the properties of materials, such as chemical composition[2]. The very minor change due to the difference in molecular structure has not been considered previously to affect charging. Hence, it is surprising that structural changes, especially due to the difference of only one atom, have significant influence over charging. Molecular structural (e.g., orientation) differences are abundantly common across surfaces. Hence, atomic-level structural differences may be the underlying reason that gives rise to the highly sensitive nature of contact electrification. This high sensitivity may cause the numerous unexplained phenomena of contact electrification, including the mosaic pattern of charge on a surface with a uniform chemical composition[18], systematically oppositely charged surfaces by contact electrification of two chemically identical materials[51], conflicting results from different research groups[52], and highly stochastic nature of contact electrification.

## Methods

### Materials

Mica was purchased from Latech Scientific Supply Pte. Ltd.

Trimethoxymethylsilane, ethyltriethoxysilane, butyltrichlorosilane, pentyltrichlorosilane, trichloro(hexyl)silane, trichloro(octyl)silane, (3-bromopropyl) trichlorosilane (90%), and ethanol were purchased from Sigma-Aldrich. $n$-Propyltrimethoxysilane and $n$-heptyltrichlorosilane were purchased from Gelest, Inc. All chemicals were used as received. Deionized water that was ultrafiltered to 18 MΩ·cm using a Milipore Milli-Q gradient system was used in all experiments.

### Coating self-assembled monolayer (SAM) of alkylsilanes onto mica

SAM of alkylsilanes was coated onto the surface of mica via a typical procedure[35,53,54]. Pieces of mica were first treated with piranha solution

at 60 °C for 20 min. They were then immersed in a silane solution that consisted of 110 μL of silane dissolved in 10 mL of 95% ethanol for 30 min at ambient temperature. After soaking, the pieces of mica were then rinsed thoroughly with 95% ethanol three times and dried in an oven at 120 °C for 1 h.

The process of coating silanes onto the surface of mica has been studied extensively in previous studies[54–58]. The process has been proposed to involve mainly two steps. The first step involves the hydrolysis of the head groups of the silanes to the reactive silanol groups. These silanol groups then react with the silanol groups present on the surface of mica, thus allowing the silanes to be grafted onto the surface of mica[59]. At the same time, the self-condensation of the silanol groups allows the silane molecules to bond with each other on the surface of mica.

## Contact electrification of coated and uncoated mica

Before the experiments, the surfaces (i.e., uncoated mica and alkylsilane-coated mica) were cleaned by either acetone or ethanol. The materials were then discharged by immersing them into water and drying them. After discharge, the uncoated mica and alkylsilane-coated mica were brought into contact and separated repeatedly for 20 times. The force applied for contacting the two pieces of materials was measured by a weighing balance. A pressure of ~100 Pa was used in typical experiments. In the experiments that used low pressure and ultra-low pressure for the contact, pressures of ~60 Pa and ~15 Pa were used respectively. The charges of both the materials were measured using a Faraday cup connected to an electrometer (Keithley, model 6514). The humidity when conducting the contact-charging experiments was ~60%. All experiments were conducted triplicate from distinct samples.

## Analysis by atomic force microscopy (AFM)

AFM images of the surfaces of the materials were obtained using a Dimension® AFM (Bruker, USA), operated in the tapping mode. Oxide sharpened $SiN_3$ cantilevers were used with a quoted spring constant of $0.04 \, N \, m^{-1}$. Data were captured at a scan rate of 1.2 Hz. The root-mean-squared (RMS) surface roughness of the uncoated surface of mica was measured to be < 0.2 nm.

## Analysis by X-ray photoelectron spectroscopy (XPS)

Quantitative elemental analyses of the surfaces were performed using XPS. XPS spectra were recorded on a PHI-5000C ESCA system (Perkin-Elmer, USA) with Al Kα excitation radiation (1486.6 eV). The pressure in the analysis chamber was maintained at $10^{-6}$ Pa during measurement. All spectra were referenced to the C 1s hydrocarbon peak at 284.6 eV to compensate for the effect of surface charging.

## Analysis by time-of-fight secondary ion mass spectrometry (ToF-SIMS)

ToF-SIMS analysis was carried out with a TRIFT II time-of-flight secondary ion mass spectrometer (ToF-SIMS 5 iontof, PHI Nano-TOF II) equipped with a $^{69}Ga^+$ liquid-metal primary ion source. Primary ion bombardment was done by 15 keV $Ga^+$ ions with a pulsed current of 600 pA. A raster size of $1 \times 1 \, \mu m$ was scanned and at least three different spots were analyzed. The total acquisition time was fixed at 180 s. Our results by ToF-SIMS of the surface of a bare (i.e., uncoated) piece of mica (before contact electrification) showed strong peaks that are typical of mica at the values of $m/z = 23, 27, 28, 39, 41$. Hence, we removed these peaks when plotting the spectra of ToF-SIMS for clear analyses of other molecular fragments besides those typical of mica.

Analysis by ToF-SIMS was performed separately for each surface. For comparing the intensities of the peaks across the different surfaces, normalization of the intensities was needed to eliminate the systematic differences due to the different measurements performed by the ToF-SIMS for each separate analysis[60]. The intensities of the peaks were normalized according to the normalization factor $K/K_0 + Si/Si_0 + Al/Al_0$. In this normalization factor, $K_0$, $Si_0$, and $Al_0$ represent the intensities of the respective elements K, Si, and Al analyzed by ToF-SIMS for the surface (i.e., either uncoated or coated mica) before contact. $K$, $Si$, and $Al$ represent the intensities of the respective elements analyzed by ToF-SIMS for the surface after contact. We considered these three elements K, Si, and Al in the normalization factor because they are the most abundant elements in mica based on our XPS analysis besides O and C (i.e., elements that may originate from other sources) (Table S1). Dividing the intensity of each of the elements after contact by the intensity of the same element before contact and then summing up the ratios of all the three elements together allowed the contribution of each element to be weighed evenly in the normalization factor. For performing the normalization, the intensities of the peaks of interest (e.g., the OH peak at $m/z = 17$) were divided by the normalization factor for comparing the intensities of the peaks across the different surfaces.

## Measurement of contact angle

The contact angle of either water or hexadecane on the surface of interest (i.e., the surface of mica uncoated or coated with the alkylsilanes) was measured. One drop of either 5 μL of water or 10 μL of hexadecane was deposited on the surface. The static contact angle was imaged using a digital camera fitted with a macro lens. The contact angle was then measured based on the images using Photoshop CS6 (Adobe). The contact angles measured were all less than 90 °C, which are similar to previously reported results[61].

## Determining group electronegativity

We found in this study that bond cleavage happened at the Si-C bond of the alkylsilane coated on the surface of mica. The bond cleavage produced an alkyl fragment that may be transferred from the alkylsilane-coated surface to the uncoated surface during contact electrification, thus leaving the $-O_3$-Si moiety behind on the surface. Heterolytic cleavage was needed for the generation of charge by contact electrification. Hence, there was a need to determine what were the polarities of charge that the alkyl fragment and $-O_3$-Si moiety gained after the heterolytic cleavage. The tendency of a chemical group to gain either a positive or negative charge after heterolytic cleavage can be determined by the concept of the electronegativity of the chemical group (i.e., group electronegativity). Based on Sanderson's principle of electronegativity equalization, the group electronegativity of a chemical group, $X_G$, can be determined by the equation indicated as follows[62].

$$X_G = \frac{N_G + q}{\sum \left( \frac{v}{x} \right)} \quad (1)$$

In this equation, $x$ is the electronegativity of an element (i.e., 2.20 for H, 2.55 for C, 3.44 for O, 1.90 for Si, and 1.61 for Al) in its isolated state, $v$ is the number of atoms of this element in the moiety (e.g., 3 for O in $-O_3$-Si), $N_G$ is the total number of atoms in the moiety (e.g., 4 for $-O_3$-Si), and $q$ is the net charge of the moiety. Based on this equation, the group electronegativities of the eight different alkyl fragments from $n = 1$ to $n = 8$ are calculated to range from 2.27 to 2.31. On the other hand, the group electronegativity of the $-O_3$-Si moiety is 2.86. Hence, the $-O_3$-Si moiety is more electronegative than all the alkyl fragments with $n = 1$ to $n = 8$. These calculations indicated that the $-O_3$-Si moiety tends to gain a negative charge and the alkyl fragments tend to gain a positive charge during heterolytic cleavage. Even if the surface layer of mica $(Al_2(AlSi_3O_{10})(OH)_2)^-$ is taken into account, the group electronegativity of the surface layer calculated by the equation is still higher than all the alkyl fragments. In addition, Si has a higher electron affinity than C. The electron affinity of Si is 1.39 eV, whereas the electron affinity of C is

1.26 eV[63]. Hence, Si has a higher tendency to gain an electron than C during the cleavage of the bond.

## Negligible effects from ions originated from water

Ions originated from water have been hypothesized previously to be responsible for charge separation by contact electrification[15]. For understanding the contribution of ions from water, we analyzed the amount of adsorbed water on the uncoated mica by ToF-SIMS. Specifically, we analyzed the hydroxide ions $(OH^-)(m/z = 17)$, water molecules $(H_2O)(m/z = 18)$, and hydronium ions $(H_3O^+)(m/z = 19)$ on the surfaces of uncoated mica both before and after contacting against the alkylsilane-coated surfaces for all $n = 1$ to 8 (Fig. S6). In all cases, we found that the amounts of hydroxide ions, water molecules, and hydronium ions were negligible. This result strongly suggested that ions originated from water did not contribute to the charge generation on the surfaces by contact electrification in our experiments.

Besides the results from ToF-SIMS, we discuss another experimental result from our study. Surfaces of mica coated with a larger number of carbon atoms are expected to be more hydrophobic due to the hydrophobicity of carbon chains. We verified that surfaces coated with a larger number of carbon atoms were indeed more hydrophobic via our measurements of the contact angles of water on the alkylsilane-coated surfaces (Fig. S5a). A surface that is more hydrophobic adsorbs less water. Hence, if ions originated from water play a major role in charge separation, there should be less charge generated by the surfaces coated with a larger number of carbon atoms. However, we observed the opposite trend. Our results showed that the amount of charge generated by contact electrification generally increases (i.e., besides the odd-even effect) when the alkylsilane-coated mica is coated with a larger number of carbon atoms (Fig. 2 and Fig. S5a).

## Characterizing the SAM coating on the surface of mica

The uncoated and SAM-coated surfaces of mica were immersed in a PBS solution containing the fluorescent molecule, FITC-BSA (20.0 μg/mL, pH 7.4), for 1 h at room temperature. The surfaces were then washed gently by immersing them in a PBS solution (i.e., without the fluorescent molecule) three times, each time with a fresh PBS solution. After washing, the surfaces were dried under vacuum at room temperature. Fluorescence images were taken by a microscope (Eclipse TE2000, Nikon, Tokyo, Japan) equipped with a highly sensitive CCD camera (ORCA-ER, Hamamatsu Photonics, Shizuoka, Japan).

The fluorescent molecule, FITC-BSA, could only adsorb on the surfaces that were coated with the alkylsilanes. Hence, the uniformity of the coating of alkylsilanes can be analyzed via the spatial distribution of the fluorescence of the surfaces. The images showed that the fluorescence was uniformly distributed across the surface, for all the surfaces coated with the alkylsilanes but not the surfaces without the alkylsilanes. These results thus showed that the alkylsilane-coated surfaces of mica were uniformly coated with the alkylsilanes.

## Analysis by Kelvin probe force microscopy (KPFM)

KPFM analysis of the surfaces was performed using a Park NX20 AFM (Park Systems, Korea), operated in non-contact FM-KPFM mode with an AC frequency of 5 kHz. Cr/Pt coated cantilevers were used with a quoted spring constant of 3 N m$^{-1}$. Data were captured at a scan rate of 0.5 Hz. The surface potential of the uncoated surface of mica increased (i.e., became more positive) after contacting the alkylsilane-coated surface of mica.

## Reporting summary

Further information on research design is available in the Nature Portfolio Reporting Summary linked to this article.

## Data availability

The charge density, surface roughness, ToF-SIMS, XPS, contact angle, and KPFM surface potential data generated in this study are provided in the Supplementary Information/Source Data file. Source data are provided with this paper.

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

## Acknowledgements

This work was financially supported by the Ministry of Education, Singapore, under grant R-279-000-496–114, R-279-000-576–114, R-279-000–633–114, and R-279-000-638–114, iHealthTech under grant R-279-001-638-731, and the Agency for Science, Technology and Research

(A*STAR) under its AME Young Individual Research Grant Scheme (Project #A1884c0021), S.S. received all the grants.

## Author contributions

Y.F., C.K.A., Y.J., Y.S. and L.C. performed the experiments. Y.F., C.K.A., and Y.J. performed all the analyses and characterization. S.S., Y.F. and C.K.A. designed the experiments. S.S., Y.F. and C.K.A. wrote the paper. S.S. conceived the project and supervised the work.

## Competing interests

The authors declare no competing interests.
