## [Peer Review File · Nature Communications]

REVIEWER COMMENTS

Reviewer #1 (Remarks to the Author):

The article "Static Charge is an Ionic Molecular Fragment" shows direct evidence of heterolysis in contact charging of covalently bonded organic materials. It is an important contribution to the field as it is the first direct evidence of heterolysis and material transfer in contact electrification at the molecular scale. The material transfer has been demonstrated in literature before by AFM and XPS. However, these techniques cannot distinguish the transfer at the molecular scale, but macroscopic pieces transferred from contact separation may not be the origin of charging. The experiments are well-planned and performed. The results are highly convincing. The manuscript would be an excellent addition to Nature Communications.

Before acceptance, I suggest that authors provide a detailed mechanistic introduction to emphasize that many studies have been reported indirectly indicating material transfer as a sole mechanism for polymer contact electrification. For example, it has been shown that the softer [Energy Environ. Sci., 2019,12, 2417-2421] and more adhesive [Macromol. Mater. Eng. 2020, 305, 1900638] polymers tend to gain larger surface charge due to pronounced heterolysis and material transfer. It has been shown that the glassy polymers observe an order of magnitude larger surface charge upon heating and transition to a rubbery state [Mater. Horiz., 2020,7, 520-523]. The contact-electrification can be observed between chemically identical polymers with different thermal histories [Energy Environ. Sci., 2019,12, 2417-2421], roughness [Phys. Chem. Chem. Phys., 2020,22, 13299-13305], and concentration of fillers [J. Mater. Chem. A, 2021,9, 8984-8990] due to the unequal amount of material transfer. It has been found that softer and smoother polymers at the polymer-polymer contact tend to observe a negative charge, while the rougher and harder polymers tend to gain a positive surface charge [ACS Appl. Mater. Interfaces 2021, 13, 37, 44935–44947, Nano Energy 104, 2022, 107914], due to more transfer from soft/smooth to hard/rough.

Reviewer #2 (Remarks to the Author):

The authors proposed a new identity of static charge, an ionic molecular fragment, based on the contact-separation measurement of an alkylsilane-coated surface of mica. Using AFM, XPS, and ToF-SIMS measurements for alkylsilane-coated surface with odd and even numbers of carbon atoms, they claimed the exact location of heterolytic cleavage of covalent bond, exact charged species, and transfer for charge separation. The manuscript is interesting, and give a new input to the contact electrification

community and possibly to the mechanical energy harvesting community. However, there are several points to be clarified and explained more.

1. I wonder the proposed mechanism could be also applicable to other materials like polymer-metal contact, which is widely used for the mechanical energy harvesting community.
2. The authors need more explanation of how the broken molecular fragment can be attached to the counter mica substrate.
3. The authors performed the contact-charging measurements at high humidity conditions (~60%). I wonder how the author assured the negligible effects from other ions originating from H₂O.
4. The authors should present the chemical homogeneity of coated SAM of alkylsilane in mica, at least to the similar area used for contact-separation measurement.
5. More sophisticated surface-sensitive measurements is required to clarify the bond cleavage, such as KPFM measurement.

Reviewer #3 (Remarks to the Author):

The article entitled “Static Charge is an Ionic Molecular Fragment” describes a study of contact electrification of surfaces modified by alkyl-silanes. The article is well written and detailed, providing strong evidence for the presence of mass transfer and molecular fragment generation during contact electrification processes. The odd-even effect is intriguing, and the study analyzing homolytic bond cleavage position is particularly nice and useful for the field. The experimental study is detailed, well thought out, and comprehensive. While I believe the manuscript is a strong addition for Nature Communications, the manuscript needs to be updated for specificity to 1D covalent bond networks (i.e., polymers) unless further evidence can be provided for applicability to 2D or 3D insulators. Please find my detailed comments below.

1) The authors state that mass transfer is not commonly debated in literature as the source of static charge (pg 3; line 77) – however there are large bodies of literature that discuss and provide strong supporting evidence for contact electrification.[1-17] These should be integrated into the introduction to provide a broader view of the field. Notably the work of Bartosz Grzybowski provides key evidence for material transfer via the formation of charged mosaics[2, 10, 18-20] – although the role of humidity in these experiments remains an open question.

2) The authors make several sweeping statements around ‘these surfaces were representative for studying contact electrification of general types of insulating materials, especially polymers’. Whilst the

model alkyl-silanated mica surfaces are excellent models of contact electrification in polymers, it is unclear how such surfaces – with a 1D polymer-like covalent backbone – mimics the charge transfer processes that occur in insulating ceramics with 3D or 2D covalent bonding networks. The adhesion force, and ability for heterolytic bond cleavage and mechano-ion transfer to occur, is much lower for harder systems with 2D or 3D bonding networks,[1, 21] and the energy required to remove an ion from a host surface is also much higher. I would suggest such sweeping statements be removed or clarified.

3) I would suggest the title be update to clarify the above point ‘Static-Charge from Polymers is an Ionic Molecular Fragment’

4) Please provide the force/instrument used for contact separation experiments described in page 5 line 136;

5) What was the humidity at which contact-electrification experiments were conducted? This should be included as Jimidar et al. have recently demonstrated a key linking between surface charge and humidity.[22] Further, the authors highlight that humidity is required for the bond-OH detection (pg 8, line 197).

6) The odd-even effect is extremely interesting – but I cannot see a clear logic as to why this has occurred. Is there a change in the structure of the initial alkyl-silane layer at odd or even chain lengths which influences chain entanglement/intramolecular forces (and subsequent bond scission)?

7) The paper would be strengthened the by drawing an explicit correlation between either the amount of detected ions by ToF-SIMS or measured roughness with surface charge. It is implied in the paper but having it spelt out better would be advantageous.

[1] A. Šutka, et al., *Advanced Materials Interfaces* 2023, n/a, 2300323, <https://doi.org/10.1002/admi.202300323>

[2] H. T. Baytekin, et al., *Science* 2011, 333, 308, doi:10.1126/science.1201512

[3] P. Yang, et al., *Matter* 2023, 6, 1295, <https://doi.org/10.1016/j.matt.2023.02.006>

[4] P. C. Sherrell, et al., *ACS Applied Materials & Interfaces* 2021, 13, 44935, 10.1021/acsami.1c13100

[5] L. Zhang, et al., *Nano Energy* 2022, 95, 107011, <https://doi.org/10.1016/j.nanoen.2022.107011>

[6] O. Verners, et al., *Nano Energy* 2022, 104, 107914,

- [7] L. Lapčinskis, et al., *Macromolecular Materials and Engineering* 2020, 305, 1900638, <https://doi.org/10.1002/mame.201900638>
- [8] R. G. Horn, et al., *Science* 1992, 256, 362, 10.1126/science.256.5055.362
- [9] X. Xia, et al., *Nano Energy* 2020, 78, 105343, <https://doi.org/10.1016/j.nanoen.2020.105343>
- [10] Y. I. Sobolev, et al., *Nature Physics* 2022, 10.1038/s41567-022-01714-9
- [11] A. Šutka, et al., *Physical Chemistry Chemical Physics* 2020, 22, 13299, 10.1039/D0CP01947J
- [12] A. Diaz, *The Journal of Adhesion* 1998, 67, 111,
- [13] M. Kaponig, et al., *Science Advances* 2021, 7, eabg7595, doi:10.1126/sciadv.abg7595
- [14] U. G. Musa, et al., *Scientific Reports* 2018, 8, 2472, 10.1038/s41598-018-20413-1
- [15] M. Sakaguchi, et al., *Journal of Electrostatics* 2014, 72, 412, <https://doi.org/10.1016/j.elstat.2014.06.006>
- [16] A. Šutka, et al., *Energy & Environmental Science* 2019, 12, 2417, 10.1039/C9EE01078E
- [17] R. K. Pandey, et al., *The Journal of Physical Chemistry C* 2018, 122, 16154, 10.1021/acs.jpcc.8b04357
- [18] T. Mazur, et al., *Chemical Science* 2017, 8, 2025, 10.1039/C6SC02672A
- [19] H. T. Baytekin, et al., *Angew. Chem. Int. Ed.* 2012, 51, 4843, <https://doi.org/10.1002/anie.201200057>
- [20] B. Baytekin, et al., *Journal of the American Chemical Society* 2012, 134, 7223, 10.1021/ja300925h
- [21] D. J. Lacks, et al., *Nature Reviews Chemistry* 2019, 3, 465, 10.1038/s41570-019-0115-1
- [22] I. S. M. Jimidar, et al., *ACS Applied Materials & Interfaces* 2023, 10.1021/acsami.3c05729

Response to the Reviewers' comments

Reviewer #1 (Remarks to the Author):

The article "Static Charge is an Ionic Molecular Fragment" shows direct evidence of heterolysis in contact charging of covalently bonded organic materials. It is an important contribution to the field as it is the first direct evidence of heterolysis and material transfer in contact electrification at the molecular scale. The material transfer has been demonstrated in literature before by AFM and XPS. However, these techniques cannot distinguish the transfer at the molecular scale, but macroscopic pieces transferred from contact separation may not be the origin of charging. The experiments are well-planned and performed. The results are highly convincing. The manuscript would be an excellent addition to Nature Communications.

Reply: We sincerely thank the Reviewer very much for his/her comments and evaluating this manuscript.

Before acceptance, I suggest that authors provide a detailed mechanistic introduction to emphasize that many studies have been reported indirectly indicating material transfer as a sole mechanism for polymer contact electrification. For example, it has been shown that the softer [Energy Environ. Sci., 2019,12, 2417-2421] and more adhesive [Macromol. Mater. Eng. 2020, 305, 1900638] polymers tend to gain larger surface charge due to pronounced heterolysis and material transfer. It has been shown that the glassy polymers observe an order of magnitude larger surface charge upon heating and transition to a rubbery state [Mater. Horiz., 2020,7, 520-523]. The contact-electrification can be observed between chemically identical polymers with different thermal histories [Energy Environ. Sci., 2019,12, 2417-2421], roughness [Phys. Chem. Chem. Phys., 2020,22, 13299-13305], and concentration of fillers [J. Mater. Chem. A, 2021,9, 8984-8990] due to the unequal amount of material transfer. It has been found that softer and smoother polymers at the polymer-polymer contact tend to observe a negative charge, while the rougher and harder polymers tend to gain a positive surface charge [ACS Appl. Mater. Interfaces 2021, 13, 37, 44935–44947, Nano Energy 104, 2022, 107914], due to more transfer from soft/smooth to hard/rough.

Reply: We really thank the Reviewer very much for providing us with this expert comment for improving the manuscript. According to the comment by the Reviewer, we have now modified the introduction to include the references that the Reviewer discussed. Specifically, we have now included discussions about the effects of softness, adhesiveness, glass transition, thermal history, roughness, and concentration of fillers of the contacting materials on contact electrification and cited the respective references as mentioned by the Reviewer. This revised paragraph is included here as follows.

“A third mechanism that has been discussed to a much lesser extent involves the transfer of charged materials (i.e., matter defined as nanoscale or larger). Previous studies have found that materials transferred between surfaces after contact¹⁸⁻²¹. Other studies found that the amount of charge generated correlated with softness^{19,22-27}, adhesiveness^{26,28,29}, glass transition of polymers²², thermal history²³, roughness^{30,31}, and concentration of fillers²⁴ of the contacting materials. However, material transfer is generally not considered to be the mechanism of contact

electrification due to the uncertainty of the fundamental concept; in particular, it is currently unknown whether the materials that transferred from one surface to another are charged. The limited amount of different pieces of materials (i.e., nanoscale or larger) transferred may not account for the significant amount of charge generated by contact electrification³². In general, there is a severe lack of evidence that supports any of these mechanisms because the results reported are based on indirect and macroscale observations.”

We have included the list of references as pointed out by the Reviewer in this revised introduction as follows:

- 22 Šutka, A., Linarts, A., Mālnieks, K., Stiprais, K. & Lapčinskis, L. Dramatic increase in polymer triboelectrification by transition from a glassy to rubbery state. *Mater. Horiz.* **7**, 520-523 (2020).
- 23 Šutka, A. *et al.* The role of intermolecular forces in contact electrification on polymer surfaces and triboelectric nanogenerators. *Energy Environ. Sci.* **12**, 2417-2421 (2019).
- 24 Lapčinskis, L. *et al.* Triboelectrification of nanocomposites using identical polymer matrixes with different concentrations of nanoparticle fillers. *J. Mater. Chem. A* **9**, 8984-8990 (2021).
- 26 Sherrell, P. C. *et al.* Probing contact electrification: A cohesively sticky problem. *ACS Appl. Mater. Interfaces* **13**, 44935-44947 (2021).
- 28 Lapčinskis, L. *et al.* The adhesion-enhanced contact electrification and efficiency of triboelectric nanogenerators. *Macromol. Mater. Eng.* **305**, 1900638 (2020).
- 30 Šutka, A. *et al.* Contact electrification between identical polymers as the basis for triboelectric/flexoelectric materials. *Phys. Chem. Chem. Phys.* **22**, 13299-13305 (2020).
- 31 Verners, O. *et al.* Smooth polymers charge negatively: Controlling contact electrification polarity in polymers. *Nano Energy* **104**, 107914 (2022).”

Reviewer #2 (Remarks to the Author):

The authors proposed a new identity of static charge, an ionic molecular fragment, based on the contact-separation measurement of an alkylsilane-coated surface of mica. Using AFM, XPS, and ToF-SIMS measurements for alkylsilane-coated surface with odd and even numbers of carbon atoms, they claimed the exact location of heterolytic cleavage of covalent bond, exact charged species, and transfer for charge separation. The manuscript is interesting, and give a new input to the contact electrification community and possibly to the mechanical energy harvesting community.

Reply: We sincerely thank the Reviewer very much for his/her comment, and the many helpful comments for improving the manuscript.

However, there are several points to be clarified and explained more.

1. I wonder the proposed mechanism could be also applicable to other materials like polymer-metal contact, which is widely used for the mechanical energy harvesting community.

Reply: We thank the Reviewer very much for the very interesting and important comment (i.e., for the mechanical energy-harvesting community). We think that the proposed mechanism can conceptually also be applicable to polymer-metal contact. For the mechanism to work on polymer-metal contact, there are two main factors to consider: the force for heterolytic cleavage for generating the ionic molecular fragments and adhesiveness of the ionic molecular fragments onto the metallic surface for the transfer. We think that both these factors can be achieved by polymer-metal contact as well; hence, the mechanism potentially works for polymer-metal contacts. We provide more details of these two factors in the following paragraphs.

In our experiments, we used mica as the reference material for contacting the alkylsilane-coated surface. Metals are generally harder than mica. Harder materials experience less deformation when the surfaces are brought into contact with an applied force. Deformation of the surface allows the contact force to be dampened. Hence, a larger portion of the mechanical force is transferred to the surface when using hard materials for the contact than softer materials. This large mechanical force may then enable hard materials such as metals to cleave bonds better than softer materials (e.g., mica). Because of this reason, hard materials are generally used in the field of mechanochemistry (e.g., stainless steel balls in ball milling). Due to the large forces available during contacts between hard materials, local spots on the surfaces with high temperatures could be created for performing various types of reactions; importantly, most mechano-reactions involve first the breaking of bonds of molecules for initiating the reactions. Therefore, it is possible for metals to cleave bonds on polymeric surfaces during contact.

After cleaving the bonds (e.g., via heterolytic cleavage) on the polymeric surfaces, adhesion of the molecular fragments onto the metallic surface is needed for the transfer of charge. There are many examples reported in literature that describe the adsorption onto metal surfaces of many different types of molecules, such as metal ions, aromatic hydrocarbons, and larger molecules such as proteins (*Nat. Commun.* **9**, 716 (2018); *Surf. Sci. Rep.* **8**, 43-125 (1988); *Acc. Chem. Res.* **47**, 3369-3377 (2014); *Prog. Surf. Sci.* **71**, 95-146 (2003); *Biomaterials* **9**, 206-212 (1988)). In the field of catalysis, the adsorption of small molecules on the surface of the metallic catalysts is the crucial first step for the subsequent catalytic reactions to occur. Many types of small organic molecules, including alcohols, aldehydes, ketones, are known to adsorb onto metallic surfaces for the catalytic reactions (*Catal. Lett.* **72**, 167-175 (2001); *J. Phys. Chem.* **95**, 7433-7437 (1991); *Colloid Polym. Sci.* **270**, 917-926 (1992); *J. Phys. Chem. B* **101**, 7939-7951 (1997)). In addition, metals generally have strong affinity toward hydrophilic molecules; for example, the high surface energy of metals enables high wettability of water onto metallic surfaces. Hence, metallic surfaces may potentially also allow the hydrophilic charged molecular fragments produced by the heterolytic cleavage of the polymer surfaces to adhere onto them.

Therefore, it is possible for charge to transfer during polymer-metal contacts via cleavage of bonds and adhering the ionic molecular fragments onto the metallic surface. On the other hand, we understand that there is evidence for electron transfer when one of the contacting materials involved is a metal (e.g., *J. Phys. Chem. B* **109**, 20511-20515 (2005); *Adv. Mater.* **31**, 1901418 (2019)) Therefore, we think that both the transfer of electrons and ionic molecular fragments may occur simultaneously. In this study, we focus only on immobile static charge on insulating surfaces. Further studies (i.e., with a different experimental design) will be needed to understand which mechanism is operative and/or dominant when the contacting surface is a metal.

2. The authors need more explanation of how the broken molecular fragment can be attached to the counter mica substrate.

Reply: We thank the Reviewer for the great comment for improving the manuscript. The surface of mica has the ability to adsorb many different types of molecules via several types of intermolecular forces as reported in previous studies. First, electrostatic forces are usually present on the surface of mica. The surface of mica is typically negatively charged because some of the silicon atoms are often replaced by aluminum atoms (*Surf. Sci. Rep.* **71**, 367-390 (2016)). Therefore, it could attract the positively charged alkyl carbocations cleaved from the alkylsilane-coated surface. Previous studies have reported that electrostatic forces allow different types of molecules, such as DNA (*Biophys. J.* **85**, 2507-2518 (2003)) and protein molecules (*Langmuir* **18**, 5841-5850 (2002)), to adsorb on the surface of mica. Second, van der Waals forces are typically present on the surface of mica for adsorbing molecules. For example, many analyses have been conducted to measure these forces on the surface of mica via AFM. Substantial molecular interactions are found to be present between the surface of mica and AFM tip coated with different substances such as gold, silicon nitride, and paraffin (*Langmuir* **12**, 2859-2862 (1996)), and between two mica surfaces (*J. Adhes.* **3**, 307-314 (1972)). Therefore, van der Waals forces may also contribute to the adsorption of molecular fragments onto the uncoated counter mica surface as discussed in our work. In addition, after the alcohols formed on the surface of the counter mica substrate (as detected by ToF-SIMS in our work), hydrogen bonds between the oxygen atoms on the surface of mica and the hydrogen atoms from the hydroxyl group of the alcohols may allow alcohols to be adsorbed onto the mica surface. It is found in previous studies that hydrogen bonds influenced the adsorption of molecules with polar functional groups such as amine and carboxylic groups onto surfaces of mica (*J. Ind. Eng. Chem.* **56**, 342-349 (2017)).

According to the comment by the Reviewer, we have now included the explanation of how the ionic molecular fragments can be adsorbed onto the surface of mica. The paragraph is included here as follows.

“All these results enabled us to determine the elementary steps of the molecular mechanism of contact electrification. When the surfaces are brought into contact, the frictional force between the contacting surfaces causes the heterolytic cleavage of the alkylsilanes functionalized on the surface — specifically, the heterolytic cleavage occurs at the polar Si-C bond. This cleavage produces a $-O_3-Si$ moiety that remains attached onto the surface and a free alkyl fragment. Based on the Sanderson’s principle of electronegativity equalization, the group electronegativity of the $-O_3-Si$ moiety is calculated to be higher than that of the alkyl fragment (see Supporting Information for the calculation). Hence, the heterolytic cleavage produces a negatively charged $-O_3-Si$ moiety on the surface and a free alkyl carbocation. The alkyl carbocation transfers from the alkylsilane-coated surface to the uncoated surface during contact. Surfaces of mica are reported to have the ability to adsorb many different types of molecules and ions via intermolecular forces, such as van der Waals forces^{45,46} and electrostatic forces (i.e., due to the typically negatively charged surface of mica^{47,48}). After separating the surfaces, the alkylsilane-coated surface charges negatively, whereas the uncoated surface charges positively due to the presence of the alkyl carbocations. The surface roughness of the uncoated surface increases due

to the transfer of the ionic molecular fragments (i.e., alkyl carbocations). Because surfaces with n_{even} experience larger frictional forces than surfaces with n_{odd} during contact, there is more heterolytic cleavage of covalent bonds. Hence, more ionic molecular fragments are transferred and more charge is separated when the contacting surfaces had n_{even} than n_{odd} (Figure 1b and 1c). This mechanism thus gives rise to the strong correlation between charge generation and surface roughness. Importantly, this molecular mechanism indicated that the static charge generated by contact electrification is an alkyl carbocation.

The alkyl carbocation then reacts spontaneously to become an alcohol, thus leaving the H^+ ion behind as the charged species on the surface (e.g., by remaining adsorbed on the surface of mica via van der Waals forces and/or hydrogen bonds). On the other hand, alkyl radicals produced by homolytic cleavage of bonds react via many different reaction pathways to form a wide range of products under normal ambient conditions, but not alcohols unless under specific conditions⁴⁹. Previous studies that investigated reactions of alkyl radicals have not detected alcohols as the products⁵⁰. In addition, the intensities of the species that are typically involved or produced in the radical reactions (i.e., R_{n-1}CHO , $\text{R}_n\text{O}_2\bullet$, $\text{R}_n\text{O}\bullet$, R_nOOH , R_nONO_2 , R_nOONO_2 , and R_nOOR_n) are found to be mostly negligible from our ToF-SIMS analyses.”

3. The authors performed the contact-charging measurements at high humidity conditions (~60%). I wonder how the author assured the negligible effects from other ions originating from H₂O.

Reply: We thank the Reviewer again for the excellent comment. Ions originating from water (i.e., the hydroxide and hydronium ions) have been hypothesized by GM Whitesides (Harvard) in a review article (*Angew. Chem. Int. Ed.* **47**, 2188-2207 (2008)) to cause the separation of charge by contact electrification. This hypothesis has since become an important point when discussing the fundamental mechanism of contact electrification. We think that these ions originating from water may not be the underlying cause for the results observed in our experiments for the reasons described in the following paragraphs.

First, the essential requirement of this mechanism based on ions originating from water is that there must be water adsorbed on the surface. We have now analyzed the amount of adsorbed water on the uncoated substrates via ToF-SIMS. We found negligible amounts of hydroxide ions (OH^-)($m/z = 17$), water molecules (H_2O)($m/z = 18$), and hydronium ions (H_3O^+)($m/z = 19$) on the surfaces of uncoated mica both before and after contacting against the alkylsilane-coated surfaces for all $n = 1$ to 8. These results are now shown in the newly added Figure S6. This result strongly suggested that ions originating from H_2O did not contribute to the charge generation on the surfaces by contact electrification in our experiments.

Besides this result from ToF-SIMS, another experimental result from our study seems to also suggest that ions originating from water do not contribute to the charge separation in our work. We first note that because of the hydrophobicity of carbon chains, alkylsilane-coated surfaces with a larger number of carbon atoms are expected to be more hydrophobic. Indeed, our measurements of the contact angles of water on the alkylsilane-coated mica (Figure S5a) verified that surfaces coated with a larger number of carbon atoms are more hydrophobic. A surface that is more hydrophobic adsorbs less water. Hence, if ions originating from water play a major role

in charge separation, then there should be less charge generated by the surfaces coated with a larger number of carbon atoms. However, we observed the opposite trend. Our results showed that the amount of charge generated by contact electrification generally increases (i.e., besides the odd-even effect) when the alkylsilane-coated mica is coated a larger number of carbon atoms (Figure 2 and Figure S5a).

In addition, the mechanism based on ions originating from water is not widely accepted as the mechanism of contact electrification in our community. There are studies that report conflicting results. For example, a study reported that charge separation occurred even when water was not present (*Angew. Chem. Int. Ed.* **50**, 6766-6770 (2011)). Other studies reported that charge separation was not affected by humidity (*Mater. Today* **30**, 34-51 (2019); *Mater. Today* **64**, 61-71 (2023)). For our mechanism, on the other hand, the involvement of water for converting the carbocations to alcohols is not necessary for the process of charge separation.

Besides the hydroxide-hydronium ion transfer model, humidity may also play a role in other situations. When humidity is varied, surfaces are found to be charged under high humidity environment spontaneously (*J. Am. Chem. Soc.* **131**, 11381-11386 (2009); *Langmuir* **26**, 13763-13766 (2010); *Anal. Chem.* **84**, 10191-10198 (2012), *ACS Appl. Mater. Interfaces* **15**, 42004-42014 (2023)). In these studies, however, variations in the humidity of the surroundings were involved. On the other hand, the environment in our laboratory is constantly controlled at ~60%; hence, there was no significant variation in humidity in our experiments.

In response to the comment by the Reviewer, we have now provided the discussion of the negligible effects of ions originating from water in a newly added section in the Supporting Information.

“Negligible effects from ions originated from water. Ions originated from water have been hypothesized previously to be responsible for charge separation by contact electrification.¹³ For understanding the contribution of ions from water, we analyzed the amount of adsorbed water on the uncoated mica by ToF-SIMS. Specifically, we analyzed the hydroxide ions (OH^-)($m/z = 17$), water molecules (H_2O)($m/z = 18$), and hydronium ions (H_3O^+)($m/z = 19$) on the surfaces of uncoated mica both before and after contacting against the alkylsilane-coated surfaces for all $n = 1$ to 8 (Figure S6). In all cases, we found that the amounts of hydroxide ions, water molecules, and hydronium ions were negligible. This result strongly suggested that ions originated from water did not contribute to the charge generation on the surfaces by contact electrification in our experiments.

Besides the results from ToF-SIMS, we discuss another experimental result from our study. Surfaces of mica coated with a larger number of carbon atoms are expected to be more hydrophobic due to the hydrophobicity of carbon chains. We verified that surfaces coated with a larger number of carbon atoms were indeed more hydrophobic via our measurements of the contact angles of water on the alkylsilane-coated surfaces (Figure S5a). A surface that is more hydrophobic adsorbs less water. Hence, if ions originated from water play a major role in charge separation, there should be less charge generated by the surfaces coated with a larger number of carbon atoms. However, we observed the opposite trend. Our results showed that the amount of charge generated by contact electrification generally increases (i.e., besides the odd-even effect)

when the alkylsilane-coated mica is coated with a larger number of carbon atoms (Figure 2 and Figure S5a).

Figure S6. ToF-SIMS spectra of the surfaces of uncoated mica before and after contact with C_n-mica for n = 1 to n = 8. The plots show the peaks corresponding to m/z = 17 (OH⁻), 18 (H₂O), 19 (H₃O⁺) and C_nH_{2n+1}OH.”

4. The authors should present the chemical homogeneity of coated SAM of alkylsilane in mica, at least to the similar area used for contact-separation measurement.

Reply: We thank the Reviewer very much for his/her comments. In response to the Reviewer’s comment, we have now performed an experiment to analyze the chemical homogeneity of the coated SAM of alkylsilane on mica. The experiment involved adsorbing a fluorescent molecule, FITC-BSA protein, onto the uncoated mica and coated mica for n = 1 to 8 and analyzing the surfaces using fluorescence microscopy. The results showed that the surfaces were uniformly coated with the alkylsilanes. The analysis involved millimeter-scale imaging; hence, the result is representative of the macro-scale area used in our experiments.

We have now added these new results to this revised version of the manuscript. These new results are added as a new section in the Supporting Information and a new figure, Figure S7.

“Characterizing the SAM coating on the surface of mica. The uncoated and SAM-coated surfaces of mica were immersed in a PBS solution containing the fluorescent molecule, FITC-BSA (20.0 μg/mL, pH 7.4), for 1 h at room temperature. The surfaces were then washed gently by immersing them in a PBS solution (i.e., without the fluorescent molecule) three times, each time with a fresh PBS solution. After washing, the surfaces were dried under vacuum at room temperature. Fluorescence images were taken by a microscope (Eclipse TE2000, Nikon, Tokyo, Japan) equipped with a highly sensitive CCD camera (ORCA-ER, Hamamatsu Photonics, Shizuoka, Japan).

The fluorescent molecule, FITC-BSA, could only adsorb on the surfaces that were coated with the alkylsilanes. Hence, the uniformity of the coating of alkylsilanes can be analyzed via the spatial distribution of the fluorescence of the surfaces. The images showed that the fluorescence

was uniformly distributed across the surface, for all the surfaces coated with the alkylsilanes but not the surfaces without the alkylsilanes. These results thus showed that the alkylsilane-coated surfaces of mica were uniformly coated with the alkylsilanes.

Figure S7. Analysis of the surfaces of mica coated with the SAM. Fluorescence images of the uncoated mica without washing, uncoated mica with washing, and coated mica, including C1-SAM, C2-SAM, C3-SAM, C4-SAM, C5-SAM, C6-SAM, C7-SAM, C8-SAM, C8-SAM with the edge shown as a contrast.”

5. *More sophisticated surface-sensitive measurements is required to clarify the bond cleavage, such as KPFM measurement.*

Reply: We thank the Reviewer very much for his/her comments. In response to the Reviewer’s comment, we have now performed the KPFM measurements of the surfaces.

We performed many analyses of many different types of surfaces, including the uncoated and coated surfaces of mica before and after contact electrification. In addition, we used different resolutions of scanning the surfaces, including the highest resolution of as small as 5×5 nm to the lowest resolution of 1×1 μm . After analyzing the many types of surfaces, we tried different ways to characterize the surfaces, including the strength, amplitude, frequency, and shape of the signals. However, we were not able to obtain any meaningful conclusion from the analyses due to the noise of the data. One thought was that by analyzing the surfaces to an area of as small as 5×5 nm (i.e., pixels scanned should be molecular scale), we may be able to obtain molecular-level information. However, the signals produced were too noisy to have any meaningful conclusion. In general, it is not expected of KPFM to produce molecular-scale level information; hence, it is not unexpected that we did not obtain results to support the main claims of this work at the molecular level. The only consistent result that we obtained is that the amount of charge increased after contact electrification (as expected).

In response to the Reviewer's comment, we have now included this result of increased in charge after contact electrification as a new section in the Supporting Information and a newly added figure, Figure S8. The modifications are included here as follows.

“Analysis by Kelvin probe force microscopy (KPFM). KPFM analysis of the surfaces was performed using a Park NX20 AFM (Park Systems, Korea), operated in non-contact FM-KPFM mode with an AC frequency of 5 kHz. Cr/Pt coated cantilevers were used with a quoted spring constant of 3 N m^{-1} . Data were captured at a scan rate of 0.5 Hz. The surface potential of the uncoated surface of mica increased (i.e., became more positive) after contacting the alkylsilane-coated surface of mica.

Figure S8. KPFM analysis of the uncoated mica before and after contact with C₂-SAM. The results showed that the uncoated mica became more positively charged after contact. This result agrees with the measurements of charge using the Faraday cup.”

Reviewer #3 (Remarks to the Author):

The article entitled “Static Charge is an Ionic Molecular Fragment” describes a study of contact electrification of surfaces modified by alkyl-silanes. The article is well written and detailed, providing strong evidence for the presence of mass transfer and molecular fragment generation during contact electrification processes. The odd-even effect is intriguing, and the study analyzing homolytic bond cleavage position is particularly nice and useful for the field. The experimental study is detailed, well thought out, and comprehensive.

Reply: We greatly appreciate the positive comments by the Reviewer. We would also like to thank the Reviewer very much for his/her many comments for improving the manuscript.

While I believe the manuscript is a strong addition for Nature Communications, the manuscript needs to be updated for specificity to 1D covalent bond networks (i.e., polymers) unless further evidence can be provided for applicability to 2D or 3D insulators. Please find my detailed comments below.

1) The authors state that mass transfer is not commonly debated in literature as the source of static charge (pg 3; line 77) – however there are large bodies of literature that discuss and

provide strong supporting evidence for contact electrification.[1-17] These should be integrated into the introduction to provide a broader view of the field. Notably the work of Bartosz Grzybowski provides key evidence for material transfer via the formation of charged mosaics[2, 10, 18-20] – although the role of humidity in these experiments remains an open question.

Reply: We sincerely thank the Reviewer for his/her comment and for providing us with the references related to material transfer. One thing is that our group is aware of the literature related to material transfer, especially the great works of Bartosz Grzybowski. We note that the corresponding author of this present work (Soh, S.) was a Ph.D. student under the supervision of Bartosz Grzybowski at Northwestern University; hence, the academic background of the corresponding author is based on material transfer. He is also a co-author of the first work by Bartosz Grzybowski that provides the key evidence for material transfer via the formation of charged mosaics when he was a Ph.D. student in his group:

Baytekin, H. T.; Patashinski, A. Z.; Branicki, M.; Baytekin, B.; Soh, S.; Grzybowski, B. A. The Mosaic of Surface Charge in Contact Electrification. *Science* 2011, 333, 308-312.

We apologize that we gave the impression that there is no discussion in literature or evidence supporting material transfer. We meant to describe the situation that material transfer is much less discussed than electron transfer (Allen Bard, UT Austin and Zhonglin Wang, Georgia Tech) and ion transfer (GM Whitesides, Harvard), in a relative sense. Electron transfer and ion transfer have always been the two mechanisms of focus in the field of contact electrification from the early days of 1950s (and possibly earlier) to today.

In response to the Reviewer's comment, we have now cited the twelve papers that the Reviewer mentioned. The contents of the papers can be divided into a few broad categories, including the correlation between charge separation and factors such as softness, adhesiveness, glass transition of polymers, thermal history, roughness, and concentration of fillers of the materials.

In this revised version of the manuscript, we have now included this discussion and the references mentioned by the Reviewer. This revised paragraph is copied here as follows.

“A third mechanism that has been discussed to a much lesser extent involves the transfer of charged materials (i.e., matter defined as nanoscale or larger). Previous studies have found that materials transferred between surfaces after contact¹⁸⁻²¹. Other studies found that the amount of charge generated correlated with softness^{19,22-27}, adhesiveness^{26,28,29}, glass transition of polymers²², thermal history²³, roughness^{30,31}, and concentration of fillers²⁴ of the contacting materials. However, material transfer is generally not considered to be the mechanism of contact electrification due to the uncertainty of the fundamental concept; in particular, it is currently unknown whether the materials that transferred from one surface to another are charged. The limited amount of different pieces of materials (i.e., nanoscale or larger) transferred may not account for the significant amount of charge generated by contact electrification³². In general, there is a severe lack of evidence that supports any of these mechanisms because the results reported are based on indirect and macroscale observations.”

Below are the references cited as suggested by the Reviewer.

- 18 Baytekin, H. T. *et al.* The mosaic of surface charge in contact electrification. *Science* **333**, 308-312 (2011).
- 19 Baytekin, H. T., Baytekin, B., Incorvati, J. T. & Grzybowski, B. A. Material transfer and polarity reversal in contact charging. *Angew. Chem. Int. Ed.* **51**, 4843-4847 (2012).
- 20 Baytekin, B., Baytekin, H. T. & Grzybowski, B. A. What really drives chemical reactions on contact charged surfaces? *J. Am. Chem. Soc.* **134**, 7223-7226 (2012).
- 21 Sobolev, Y. I., Adamkiewicz, W., Siek, M. & Grzybowski, B. A. Charge mosaics on contact-electrified dielectrics result from polarity-inverting discharges. *Nat. Phys.* **18**, 1347-1355 (2022).
- 23 Šutka, A. *et al.* The role of intermolecular forces in contact electrification on polymer surfaces and triboelectric nanogenerators. *Energy Environ. Sci.* **12**, 2417-2421 (2019).
- 25 Pandey, R. K., Kakehashi, H., Nakanishi, H. & Soh, S. Correlating material transfer and charge transfer in contact electrification. *J. Phys. Chem. C* **122**, 16154-16160 (2018).
- 26 Sherrell, P. C. *et al.* Probing contact electrification: A cohesively sticky problem. *ACS Appl. Mater. Interfaces* **13**, 44935-44947 (2021).
- 27 Šutka, A. *et al.* Engineering polymer interfaces: A review toward controlling triboelectric surface charge. *Adv. Mater. Interfaces* **10**, 2300323 (2023).
- 28 Lapčinskis, L. *et al.* The adhesion-enhanced contact electrification and efficiency of triboelectric nanogenerators. *Macromol. Mater. Eng.* **305**, 1900638 (2020).
- 29 Zhang, L. *et al.* Mechanism and regulation of peeling-electrification in adhesive interface. *Nano Energy* **95**, 107011 (2022).
- 30 Šutka, A. *et al.* Contact electrification between identical polymers as the basis for triboelectric/flexoelectric materials. *Phys. Chem. Chem. Phys.* **22**, 13299-13305 (2020).
- 31 Verners, O. *et al.* Smooth polymers charge negatively: Controlling contact electrification polarity in polymers. *Nano Energy* **104**, 107914 (2022).

2) The authors make several sweeping statements around ‘these surfaces were representative for studying contact electrification of general types of insulating materials, especially polymers’. Whilst the model alkyl-silanated mica surfaces are excellent models of contact electrification in polymers, it is unclear how such surfaces – with a 1D polymer-like covalent backbone – mimics the charge transfer processes that occur in insulating ceramics with 3D or 2D covalent bonding networks. The adhesion force, and ability for heterolytic bond cleavage and mechano-ion transfer to occur, is much lower for harder systems with 2D or 3D bonding networks,[1, 21] and the energy required to remove an ion from a host surface is also much higher. I would suggest such sweeping statements be removed or clarified.

3) I would suggest the title be update to clarify the above point ‘Static-Charge from Polymers is an Ionic Molecular Fragment’

Reply: The Reviewer mentioned that we should change the discussion of our work to include only polymers and not general types of insulating materials, such as ceramics. While we see the perspective of the Reviewer, we think that there are several strong reasons conceptually to think that the mechanism of our study is general, including for ceramics.

The Reviewer is certainly right that ceramics are generally harder than polymers and are unfavorable to separate charge by contact electrification. A very important point is that ceramics generally charge lower — by one to two orders of magnitude — by contact electrification than polymers. This phenomenon is commonly observed, including in many previous studies (*Nat. Commun.* **11**, 2093 (2020); *Nat. Commun.* **10**, 1427 (2019)) and also experience from our laboratory. Hence, there is a correlation between the unfavorable aspects of ceramics for contact electrification than polymers (i.e., as stated by the Reviewer) versus the amount of charge generated. This commonly observed phenomenon thus bridges the two aspects: it seems to us that *both* the Reviewer is right, *and* our proposed mechanism is also possibly applicable to ceramics.

There are other reasons too. For this discussion, we will reply to the two points that the Reviewer mentioned separately: (1) the ability to undergo heterolytic bond cleavage and (2) adhesion force for mechano-ion transfer to occur.

We first compare (1) the ability of ceramics versus polymers to undergo heterolytic bond cleavage. An important point is that the number of covalent bonds is fundamentally limited by the chemistry of the atoms, regardless of whether the structure is 1D, 2D, or 3D. A common example of ceramics is quartz (i.e., crystalized SiO₂); the structure of quartz involves Si-O-Si bonds. Due to the chemistry of oxygen, the oxygen atoms are limited to only 2 covalent bonds, even when they are within the 3D bulk ceramic. This type of bonding is similar to polymers *at the molecular scale*: polymers (e.g., polyethylene) involve chains of C-C bonds where each carbon atom is commonly also limited to only 2 covalent bonds with other atoms (if we disregard hydrogen atoms). Hence, our proposed molecular mechanism may also apply to these types of ceramics. Other atoms in ceramics may have more than 2 covalent bonds, thus causing these ceramics to charge lower by contact electrification.

Another important factor is that contact electrification is a surface phenomenon — hence, the material structure to be considered inevitably involves 2D surfaces. Importantly, a 2D surface is a truncation of the 3D bulk material — the lack of the third dimension typically causes atoms on the surface to be less bonded. As a result, high densities of molecular groups have only one (i.e., similar to our case of 1D covalent bonding) or two covalent bonds bonded onto the surface, including on surfaces of ceramics. Hence, these molecular groups that are less bonded may potentially undergo the same mechanism as described in this study.

A specific example involves the general class of oxides (i.e., a type of ceramics). Oxides typically have terminal hydroxyl groups on their surfaces that are each connected by only one covalent bond onto the surface (i.e., due to reaction with atmospheric moisture) (*J. Mol. Struct.* **19**, 579-589 (1973); *Phys. Rev. B* **76**, 205415 (2007); *Phys. Rev. B* **78**, 045416 (2008); *Prog. Surf. Sci.* **85** 161-205 (2010); *Langmuir* **37**, 10588-10593 (2021); *J. Phys. Chem.* **99**, 4639-4647 (1995)). The surface coverages of the singly bonded -OH groups are found to be on the order of 1-10 nm⁻² on many different oxides, including SiO₂, Al₂O₃, Fe₂O₃, TiO₂, Cr₂O₃, and MnO₂ (*J. Phys. Chem. C* **120**, 21427-21435 (2016), *Langmuir* **3**, 316-318 (1987); *J. Phys. Chem.* **99**, 4639-4647 (1995); *Surf. Interface Anal.* **26**, 549-564 (1998); *J. Colloid Interface Sci.* **209**, 225-231 (1999)). Therefore, the typical charge densities on the order of 1 μC/m² generated by contact electrification can easily be obtained by cleaving only a very small fraction (i.e., 1/100000) of

the surface -OH groups during contact. Oxide layers are also commonly found on surfaces of non-oxide ceramics (*Nat. Rev. Chem.* **3**, 465-476 (2019)); thus, the same mechanism may occur. Besides terminal -OH groups, high densities of many other types of functional groups bonded with only one covalent bond onto the surface are often found on ceramics, including Si-CH₃ groups (*J. Am. Ceram. Soc.* **72**, 1692-1697 (1989)) and primary amine groups on surfaces of boron nitride (*Surf. Interface Anal.* **37**, 621-627 (2005)).

Another important point is that in the field of mechanochemistry, hard materials, including insulating ceramics, are usually used for the purpose of breaking bonds (i.e., instead of polymers). In a typical experiment involving ball milling, the balls are typically made of hard materials, including ceramics, for performing the mechano-reactions. Hard materials are used because they do not deform like soft materials; hence, they tend to transfer the mechanical forces applied onto them to break chemical bonds effectively. This effective transfer of the mechanical forces tends to create local heating to a very high temperature during contact (e.g., at the asperities). The local heating may cause a rise in temperature of a few hundreds to thousands of degrees at local spots, thus leading to breaking of bonds for performing the chemical reactions (*Chem. Soc. Rev.* **42**, 7719-7738 (2013); *Chem. Soc. Rev.* **42**, 7649-7659 (2013)). Hence, hard materials have the advantage of having the force for cleaving the bonds of molecules on surfaces.

On the other hand, some ceramics are formed by weak types of covalent bonds (i.e., bond energies at 200 kJ/mol or weaker); some examples include Pb-O in PbO₂, Zn-O, Cd-O, Mo-Cl in MoCl₆, and Hg-Cl in Hg₂Cl₂ (*Comprehensive handbook of chemical bond energies*. (CRC Press, 2007)). Hence, the bonds of these ceramics can be cleaved easily; thus, we cannot rule out this class of ceramics that may potentially undergo the mechanism described in this study.

The second point (2) involves the discussion of adhesion for adsorbing molecular fragments onto the surface of ceramics. There are many studies in literature that reported the adsorption of many types of molecules and ions onto the surface of ceramics, such as metal ions, small organic molecules (e.g., aldehyde, alcohol, toluene), and simple gas molecules (e.g., O₂, N₂, CO). This ability to adsorb molecules and ions onto the surfaces is common and important; thus, it has led to many types of applications of ceramics, including sensors, membranes, and catalysts (*Langmuir* **12**, 4190-4196 (1996); *Appl. Catal., A* **167**, 195-202 (1998); *Sensors* **6**, 1751-1764 (2006); *Sep. Purif. Technol.* **49**, 49-55 (2006); *J. Colloid Interface Sci.* **348**, 579-584 (2010); *J. Photochem. Photobiol., A* **98**, 79-86 (1996); *Microporous Mesoporous Mater.* **145**, 51-58 (2011); *J. Membr. Sci.* **196**, 69-77 (2002); *J. Colloid Interface Sci.* **353**, 512-518 (2011)). Therefore, it is possible that the ionic molecular fragments generated during contact electrification also adsorb onto the surface of the opposite contacting ceramic material after contact.

Besides the investigation of the adsorption of molecules and ions, many studies measured the areal work of adhesion between pairs of materials, which is defined as the work required to separate two surfaces from each other. The areal work of adhesion is a good measure of the strength of adhesiveness between two contacting materials. From many studies conducted by many research groups using different methods of analysis (e.g., AFM or surface force apparatus), it is found that the typical values of the areal work of adhesion between two types of polymers (e.g., PS against PS or PTFE against PMMA) and between polymers and ceramics (e.g., PS against mica or PEG against graphene oxide) are similar and are both on the order of 10 to 100

mJ/m² (*Macromolecules* **49**, 5223-5231 (2016); *Wear* **84**, 167-181 (1983); *J. Mech. Phys. Solids* **156**, 104578 (2021); *J. Rheol.* **42**, 795-812 (1998); *Dent. Mater.* **20**, 338-344 (2004)). On the other hand, the reported values of the areal work of adhesion from many studies between ceramics and between ceramics and metals (e.g., graphene against Cu or SiO₂ against mica) are found to be higher and on the order of 100 to 1000 mJ/m² (*Acta Mater.* **50** 441-466 (2002); *Nanoscale* **7**, 10760-10766 (2015); *Science* **256**, 362-364 (1992)). Hence, many studies reported that ceramics actually have better adhesive forces than polymers.

Adhesion force at the macro-scale is in general a complex topic. Some polymers may be very adhesive due to other factors (e.g., conformal contact for a larger area of contact and polymeric chain entanglement) for adhering macro-scale materials. At the molecular scale, however, the results from previous studies as discussed above indicated that ceramics have high tendencies to adsorb molecules.

Further studies with different experimental designs will be needed to investigate the molecular mechanism of contact electrification of ceramics. On the other hand, based on the discussion above, it is possible that at least a substantial portion of ceramics has the potential to undergo the proposed mechanism of this study (i.e., heterolytic cleavage for generating the ionic molecular fragment and transfer for charge separation).

In response to the comment by the Reviewer, we have now improved our discussion of the generality of our study in the conclusion of this revised manuscript as follows.

“This mechanism is applicable to general types of insulating materials. Most insulating materials are covalently bonded. Hence, similar cleavage of covalent bonds will occur on the surfaces of other types of insulating materials according to the mechanism described in this study (e.g., similar cleavage of carbon-heteroatom or carbon-carbon bonds on polymeric surfaces and covalent bonds in inorganic materials). An important consideration is that surfaces are truncations of bulk 3D materials. Hence, surfaces have molecular groups that are largely less bonded covalently than within the bulk and typically have a vast variety of functionalization that produces groups that are less bonded (e.g., including groups with only one covalent bond on the surface) than within the bulk (e.g., the spontaneous functionalization of -OH groups by atmospheric moisture on inorganic materials, such as oxides). In our case, we found that the static charge is an alkyl carbocation. According to the elementary steps of this mechanism, the actual charged species generated by other types of insulating materials will depend on the types of chemical groups present on the surface — in general, static charge on insulating materials is an ionic molecular fragment.”

4) Please provide the force/instrument used for contact separation experiments described in page 5 line 136;

Reply: We thank the Reviewer very much for the comment. A pressure of ~100 Pa was used in typical experiments. In the contact-separation experiments with low pressure and ultra-low pressure, pressures of ~60 Pa and ~15 Pa were used respectively. The instrument used for measuring the pressure was a weighing balance.

In response to the Reviewer's comment, we have now included these details of the contact-separation experiment in the Supporting Information and main text.

In the Supporting Information:

“Contact electrification of coated and uncoated mica. Before the experiments, the surfaces (i.e., uncoated mica and alkylsilane-coated mica) were cleaned by either acetone or ethanol. The materials were then discharged by immersing them into water and drying them. After discharge, the uncoated mica and alkylsilane-coated mica were brought into contact and separated repeatedly for 20 times. The force applied for contacting the two pieces of materials was measured by a weighing balance. A pressure of ~100 Pa was used in typical experiments. In the experiments that used low pressure and ultra-low pressure for the contact, pressures of ~60 Pa and ~15 Pa were used respectively. The charges of both the materials were measured using a Faraday cup connected to an electrometer (Keithley, model 6514). The humidity when conducting the contact-charging experiments was ~60%.”

In the main text:

“Results showed that the amount of charge generated on both surfaces increased, in general, with increasing n (Figure 2). The increase was large. The amount of charge generated when $n = 8$ compared to $n = 1$ was around eleven times for the alkylsilane-coated surfaces and six times for the uncoated surfaces when a typical pressure of ~100 Pa was used for contacting the surfaces (“Normal pressure” in Figure 2a). Importantly, the increase was not monotonic and followed a remarkable trend: both the contacting surfaces charged higher when the alkylsilanes coated onto the surfaces had even numbers of carbon atoms, n_{even} , than odd numbers of carbon atoms, n_{odd} (i.e., an odd-even effect). For example, the negative charge of the alkylsilane-coated surface and positive charge of the uncoated surface were larger when $n = 2$ compared to when $n = 1$ or $n = 3$. This significant and systematic odd-even effect was not initially expected to occur due to the highly stochastic and complex nature of contact electrification. The phenomenon was general. Similar trends were observed when lower amounts of pressure were used for contacting the materials (“Low pressure” of ~60 Pa and “Ultra-low pressure” of ~15 Pa in Figure 2a).”

5) What was the humidity at which contact-electrification experiments were conducted? This should be included as Jimidar et al. have recently demonstrated a key linking between surface charge and humidity.[22] Further, the authors highlight that humidity is required for the bond-OH detection (pg 8, line 197).

Reply: We thank the Reviewer for his/her comment for improving the manuscript. Indeed, humidity is a crucial factor in experiments of contact electrification, especially in our case where we detected the alcohols. We previously only stated the humidity (i.e., 60%) that we used in the methods section of the Supporting Information.

In response to the comment by the Reviewer, we have now added the information about the humidity in two locations to the main text. The modifications are shown in yellow as follows.

“We performed the contact electrification by bringing the alkylsilane-coated surface of a specific n (i.e., C_n -SAM) into contact with the uncoated surface of mica (i.e., C_n -mica) twenty times under ambient conditions (i.e., humidity ~60%).”

“Alcohol is the indicator of the formation of alkyl carbocation. Alkyl carbocations react spontaneously and rapidly with water due to the humidity (i.e., ~60%) of the surrounding atmosphere under normal ambient conditions to form alcohols and H^+ ions (e.g., the essential step in the classic reaction of hydration of alkenes; Figure 3a).²⁵⁻²⁷”

6) The odd-even effect is extremely interesting – but I cannot see a clear logic as to why this has occurred. Is there a change in the structure of the initial alkyl-silane layer at odd or even chain lengths which influences chain entanglement/intramolecular forces (and subsequent bond scission)?

Reply: We thank the Reviewer again for the excellent comment. We think that the odd-even effect is fascinating and did not initially expect that it would happen since experiments of contact electrifications always include a substantial amount of noise. The Reviewer is right that the discussion of the mechanism of the odd-even effect can be clearer. There are studies conducted previously that investigated specifically the differences in the molecular structure between odd and even numbers of carbon atoms in monolayers of molecules on surfaces (although no studies have yet reported the odd-even effects of charging by contact electrification). Based on the results by these previous studies, it is possible to obtain a clear logical and complete explanation of the odd-even effect of contact electrification.

Specifically, the complete sequence of logic is as follows.

- For self-assembled monolayers (SAMs) of alkylsilanes with either an odd or even number of carbon atoms, previous studies have found that the terminal CH_3-CH_2- moiety at the topmost portion of the SAM layer can either be tilted away from the normal or directed toward the normal.
- These molecular orientations, as discussed in previous studies, can be known by measuring the odd-even contact angles of liquids on the surfaces (i.e., lower contact angles indicate that the terminal CH_3-CH_2- moiety is tilted away from the normal than directed toward the normal). We thus measured the contact angles of water and hexadecane on substrates covered with SAMs of different lengths of carbon atoms, n . By comparing our experimental results with those from the previous studies, we found that n_{even} has the terminal CH_3-CH_2- moiety tilted away from the normal and n_{odd} has the terminal CH_3-CH_2- moiety directed toward the normal.
- When the CH_3-CH_2- moiety is tilted away from the normal, both the methyl groups and methylene groups are exposed. On the other hand, when the CH_3-CH_2- moiety is directed to the normal, only the methyl groups are exposed at the topmost layer.
- When both the methyl groups and methylene groups are present on the topmost surface, there are more atoms per unit area in contact between the surfaces than when only the methyl groups are present.
- When more atoms per unit area are in contact, the surfaces experience larger adhesion forces, which lead to a larger frictional force between the surfaces.

- A larger frictional force during contact causes more heterolytic cleavage of covalent bonds.
- Hence, n_{even} has more ionic molecular fragments generated, and more charge separated than n_{odd} .

In response to the comment by the Reviewer, we have now included a more detailed explanation of the odd-even effect of contact electrification in the main text. The modified paragraphs are as follows.

“For the odd-even effect, previous studies that examined the molecular structure of the self-assembled monolayer of alkyl chains found that the orientation of the terminal CH₃-CH₂- moiety at the outermost portion of the molecule can be normal or tilted away from the normal to the surface depending on whether the molecule is n_{even} or n_{odd} . When the terminal CH₃-CH₂- moiety is normal to the surface, the outermost portion of the surface is composed of mainly the methyl groups; when the CH₃-CH₂- moiety is tilted away from the normal, the outermost portion is composed of both the methyl groups and methylene groups. This odd-even molecular orientation can be determined by analyzing the wettability of liquids on the surfaces³⁵. Specifically, the contact angles of different types of liquids (e.g., n -hexadecane and water) are lower when the terminal CH₃-CH₂- moiety is tilted away from the normal than when it is normal to the surface^{42,43}. We measured the contact angles of liquids on the alkylsilane-coated surfaces. Results showed that the contact angles of water (Figure S5a) and hexadecane for $n < 5$ (Figure S5b) were generally lower for n_{even} than for n_{odd} . This result thus indicated that the terminal CH₃-CH₂- moiety was tilted away from the normal for n_{even} and directed toward the normal for n_{odd} as illustrated in Figure 1a-c.

Previous studies have reported that the frictional force between the contacting surfaces is larger when the terminal CH₃-CH₂- moiety of the self-assembled monolayer is tilted away from the normal⁴⁴. The reason is because the presence of both the methyl groups and methylene groups when the terminal CH₃-CH₂- moiety is tilted away from the normal allows more atoms per unit area to be in contact between the surfaces than when only the methyl groups are present. When more atoms per unit area are in contact, the surfaces experience larger Van der Waals forces and a larger frictional force. In our case, this result indicates that surfaces with n_{even} experienced a larger frictional force than surfaces with n_{odd} . Hence, surfaces with n_{even} have a larger amount of cleavage of bonds than surfaces with n_{odd} .

7) The paper would be strengthened by drawing an explicit correlation between either the amount of detected ions by ToF-SIMS or measured roughness with surface charge. It is implied in the paper but having it spelt out better would be advantageous.

Reply: We thank the Reviewer very much again for the great comment. We agree with the Reviewer that it would be clearer to explicitly state the correlations in the manuscript. In response to the comment by the Reviewer, we have now more explicitly stated the correlations in the conclusion of this revised version of the manuscript. The modification is shown as follows.

“The clarity at the molecular level indicated clearly that contact electrification occurs by first the heterolytic cleavage of covalent bonds and then transfer of the ionic molecular fragments.

Besides showing that it occurs, we showed that this specific mechanism occurs in abundance — at quantities that correspond to the substantial amount of charge separation generated by contact electrification. Three results showed that the mechanism occurred in abundance. First, there is a strong correlation between the amount of charge generated (Figure 2a) and surface roughness (Figure 2b) (i.e., both showed the general increasing trend and signature odd-even effect); changes in surface roughness are indicative of large-scale transfer of molecules. Second, there is a correlation between the amount of charge generated (Figure 2a) and the transfer of molecules detected by ToF-SIMS (Figure 3e) (i.e., both showed the unique odd-even effect). Third, large amounts of transfer of alcohols are detected by ToF-SIMS (Figure 3). Previous studies have only considered the transfer of other types of species, such as electrons, mobile ions, and materials (i.e., nanoscale or larger), but not an ionic molecular fragment.”

- [1] A. Šutka, et al., *Advanced Materials Interfaces* 2023, n/a, 2300323, <https://doi.org/10.1002/admi.202300323>
- [2] H. T. Baytekin, et al., *Science* 2011, 333, 308, doi:10.1126/science.1201512
- [3] P. Yang, et al., *Matter* 2023, 6, 1295, <https://doi.org/10.1016/j.matt.2023.02.006>
- [4] P. C. Sherrell, et al., *ACS Applied Materials & Interfaces* 2021, 13, 44935, 10.1021/acscami.1c13100
- [5] L. Zhang, et al., *Nano Energy* 2022, 95, 107011, <https://doi.org/10.1016/j.nanoen.2022.107011>
- [6] O. Verners, et al., *Nano Energy* 2022, 104, 107914,
- [7] L. Lapčinskis, et al., *Macromolecular Materials and Engineering* 2020, 305, 1900638, <https://doi.org/10.1002/mame.201900638>
- [8] R. G. Horn, et al., *Science* 1992, 256, 362, 10.1126/science.256.5055.362
- [9] X. Xia, et al., *Nano Energy* 2020, 78, 105343, <https://doi.org/10.1016/j.nanoen.2020.105343>
- [10] Y. I. Sobolev, et al., *Nature Physics* 2022, 10.1038/s41567-022-01714-9
- [11] A. Šutka, et al., *Physical Chemistry Chemical Physics* 2020, 22, 13299, 10.1039/D0CP01947J
- [12] A. Diaz, *The Journal of Adhesion* 1998, 67, 111,
- [13] M. Kaponig, et al., *Science Advances* 2021, 7, eabg7595, doi:10.1126/sciadv.abg7595
- [14] U. G. Musa, et al., *Scientific Reports* 2018, 8, 2472, 10.1038/s41598-018-20413-1
- [15] M. Sakaguchi, et al., *Journal of Electrostatics* 2014, 72, 412, <https://doi.org/10.1016/j.elstat.2014.06.006>
- [16] A. Šutka, et al., *Energy & Environmental Science* 2019, 12, 2417, 10.1039/C9EE01078E
- [17] R. K. Pandey, et al., *The Journal of Physical Chemistry C* 2018, 122, 16154, 10.1021/acs.jpcc.8b04357
- [18] T. Mazur, et al., *Chemical Science* 2017, 8, 2025, 10.1039/C6SC02672A
- [19] H. T. Baytekin, et al., *Angew. Chem. Int. Ed.* 2012, 51, 4843, <https://doi.org/10.1002/anie.201200057>
- [20] B. Baytekin, et al., *Journal of the American Chemical Society* 2012, 134, 7223, 10.1021/ja300925h
- [21] D. J. Lacks, et al., *Nature Reviews Chemistry* 2019, 3, 465, 10.1038/s41570-019-0115-1
- [22] I. S. M. Jimidar, et al., *ACS Applied Materials & Interfaces* 2023, 10.1021/acscami.3c05729

Reply: We thank the Reviewer very much for providing us with the references.

REVIEWERS' COMMENTS

Reviewer #1 (Remarks to the Author):

The authors have addressed all comments properly, and the manuscript should be accepted for publishing

Reviewer #2 (Remarks to the Author):

The authors have satisfactorily addressed all of my concerns and comments. Thus, I recommend the revised manuscript for publication in Nature Communications. This research may offer a crucial insight into comprehending contact electrification phenomena.

Reviewer #3 (Remarks to the Author):

The authors have now rigorously addressed the concerns/comments of the manuscript, and made comprehensive changes. The paper is now acceptable for publication.